# Text Modality Oriented Image Feature Extraction for Detecting Diffusion-based DeepFake

## Abstract

The widespread use of diffusion methods enables the creation of highly realistic images on demand, thereby posing significant risks to the integrity and safety of online information and highlighting the necessity of DeepFake detection. Our analysis of features extracted by traditional image encoders across ten diffusion types reveals that both low-level and high-level features offer distinct advantages in identifying DeepFake images. Furthermore, the highly realistic images generated by diffusion models make it increasingly difficult to distinguish between real and fake within the image domain. Building on these insights, we propose the development of an effective representation beyond the image domain, capable of capturing both low-level and high-level features for detecting diffusion-based DeepFakes. Specifically, for a given target image, the representation we discovered is a corresponding text embedding that can guide the generation of the target image with a specific text-to-image model. Experiments conducted across ten diffusion types compared with five representative deepfake detection baselines demonstrate the efficacy of our proposed method.

## 1 Introduction

With the popularization and development of diffusion technology Croitoru et al. (2023), recent advancements in the field of synthetic content generation Ho et al. (2020); Song et al. (2020); Zhang et al. (2023); Hertz et al. (2023) have marked a new era of generative AI. Recent advancements in diffusion models enable the creation of highly realistic images that are often indistinguishable from real ones by humans Rombach et al. (2022), potentially disrupting digital media communication and introducing new societal risks york times (2024); Schick (2020). Additionally, the DeepFake detectors designed to identify fake images generated by Generative Adversarial Networks (GAN) Goodfellow et al. (2020), are not very effective against those generated by diffusion models Corvi et al. (2023); Ojha et al. (2023). Consequently, developing reliable DeepFake detection Juefei-Xu et al. (2022) methods that can keep pace with the latest generative models (*i.e.*, diffusion-based models) is extremely necessary and challenging.

For DeepFake detection, **extracting features from real and fake images for further classification is a fundamental and critical step**, as effective and deliberate feature extraction significantly benefits the performance of downstream tasks Yan et al. (2023). In terms of feature extraction, existing detection methods Wang et al. (2023); Sha et al. (2023); Ricker et al. (2024); Ojha et al. (2023) designed for diffusion-based generative models all trapped in a conventional mindset, treating the detection task as a traditional binary classification problem, similar to image classification. Specifically, these methods primarily focus on leveraging **high-level features** Zeiler (2014) (*i.e.*, extracted in deeper model layers, representing abstract patterns like object shapes and semantics, crucial for recognition tasks). In contrast, **low-level features** Zeiler (2014) (*i.e.*, extracted in the initial model layers, representing basic visual elements like edges, textures, and colors) are particularly effective in identifying generative model artifacts but are often overlooked in current diffusion-based deepfake detection tasks.

However, through our empirical study analyzing the features extracted by the image encoders of two classical model architectures, CNN He et al. (2016) and ViT Dosovitskiy et al. (2021) on a large number of DeepFake types Dhariwal & Nichol (2021); Ho et al. (2020); Nichol & Dhariwal (2021); Liu et al. (2022); Gu et al. (2022). We find although the high-level features are effective in

some diffusion types, low-level features perform better in others when distinguishing real from fake images. This observation emphasizes the need to move beyond traditional binary classification tasks in DeepFake detection and adopt a more comprehensive analysis. Specifically, careful analysis of low-level features is essential, as they can provide significant insights into an image's authenticity Bayar & Stamm (2018); Corvi et al. (2023). To summarize, **relying solely on either high-level or low-level features to construct a DeepFake detection method is not comprehensive enough. We suggest finding a representation that effectively captures both feature types.**

On the other hand, due to the highly realistic synthetic images generated by diffusion models that closely approximate the appearance of real images, distinguishing between real and generated images within the *image domain* is becoming increasingly challenging Wang et al. (2023). Therefore, exploring the possibility of extracting image information into other domains to enhance the distinguishability between real and fake images is necessary. This approach has proven effective in previous detection methods for GAN-based DeepFakes Frank et al. (2020); Zhao et al. (2021); Guo et al. (2023); Jeong et al. (2022). For instance, GAN-based DeepFakes often exhibit specific flaws in the frequency domain Frank et al. (2020); Zhao et al. (2021), such as abnormal frequency distributions or unnatural spectral features, which serve as key indicators for distinguishing between real and fake images. However, diffusion-based fake images closely resemble real ones in the frequency domain, making frequency-oriented feature extraction methods less effective Corvi et al. (2023). Therefore, developing techniques to **transform image information into another domain** is a promising approach, as these domains may expose subtle, hard-to-detect flaws in images generated by diffusion models.

Taking into account the two aforementioned considerations, we propose an approach inspired by Vision Language Models (VLMs) to create a representation that integrates both high-level and low-level features by transforming image information into the text domain. Text serves as an effective semantic starting point by capturing high-level concepts, but its inherent ambiguity means that a single prompt can yield many visually distinct images, necessitating a refinement process to incorporate low-level visual features. Therefore, we contend that a comprehensive representation must resolve this ambiguity, enabling it to steer a generative process toward a single predetermined target image. To achieve this granular control, we refine continuous text embeddings rather than discrete tokens.

Motivated by this observation, we propose a novel **T**ext modality-**O**riented **F**eature **E**xtraction method, termed **TOFE**. To be specific, given a target real or fake image, TOFE first obtains the embedding of a text input and then iteratively optimizes the embedding to be a representation that can guide the generation of the target image with a pre-trained text-to-image model. Through our experiment, we find the representation extracted by TOFE shows better performance in distinguishing real and fake images than features extracted from image encoders of classical architecture (*i.e.*, ResNet and CLIP). To summarize, our work has the following contributions:

- We explore extracting image features into the text modality for DeepFake detection, highlighting the potential of cross-modal feature extraction.

- We demonstrate that both high-level and low-level features are necessary for reliably detecting diverse diffusion-based DeepFakes, rather than relying on only one feature type.

- We conduct extensive experiments on ten diffusion types with five advanced detectors, which verify the effectiveness of the proposed representation.

## 2 RELATED WORK

### 2.1 DIFFUSION-BASED FAKE IMAGE GENERATION

In recent years, diffusion models have emerged as a mainstream approach for AI-generated content (AIGC) Zhang et al. (2023); Hertz et al. (2023); Ruiz et al. (2023); Van Le et al. (2023). Ho et al. introduced denoising diffusion probabilistic models (DDPMs) Ho et al. (2020), which achieve competitive performance compared to PGGAN Karras et al. (2018). DDPMs generate high-quality images through an iterative process of adding and then removing noise, training a neural network to progressively denoise the image. Subsequent works focus on improving architectures and sampling efficiency. The denoising diffusion implicit model (DDIM) Song et al. (2021) leverages non-Markovian processes to reduce sampling steps while preserving image fidelity. ADM Dhariwal & Nichol (2021)

refines the diffusion architecture to enhance both visual quality and training efficiency. PNDM Liu et al. (2022) further improves sampling efficiency and image clarity using pseudo-numerical methods.

To introduce controllability, classifier-free guidance Ho & Salimans (2021) allows balancing fidelity and diversity during generation. VQ-Diffusion Gu et al. (2022) integrates VQ-VAE van den Oord et al. (2017) to better handle complex scenes and improve image quality. Latent Diffusion Models (LDMs) Rombach et al. (2022) perform denoising in a compressed latent space before decoding to high-dimensional images, significantly reducing computational cost and accelerating generation. LDMs have been widely adopted, with Stable Diffusion stability ai (2024) being a prominent example.

## 2.2 DETECTION ON DIFFUSION-BASED DEEPFAKE

There is a lot of work Wang et al. (2020); Han et al. (2023); Yu et al. (2019); Huang et al. (2022) paying attention to the detection of GAN-based DeepFakes. For example, CNNDet Wang et al. (2020) shows that a detector trained on one CNN generator (*i.e.*, ProGAN) can generalize to other CNN architectures, suggesting common flaws in CNN-generated images. However, diffusion models differ fundamentally from GANs, rendering many GAN-based detectors ineffective. To address this, DIRE Wang et al. (2023) leverages the observation that diffusion-generated images can be reconstructed more accurately by pre-trained diffusion models than real images. UFD Ojha et al. (2023) uses CLIP feature space for nearest-neighbor classification, DMDet Corvi et al. (2023) studies the challenges of distinguishing diffusion-generated images, and AEROBLADE Ricker et al. (2024) employs autoencoder reconstruction error in a training-free setup. Most diffusion-based detectors focus on high-level CNN or ViT features, often neglecting low-level features, highlighting a gap in existing approaches for comprehensive feature-based detection.

## 3 FEATURE ANALYSIS FOR DEEPFAKE DETECTION

We conduct a feature analysis on the DIRE dataset Wang et al. (2023), which contains DeepFake images generated by ten diffusion models. Given that CNN-based image encoders are widely used across various tasks He et al. (2016), and that features extracted from ViT-based foundation models have demonstrated strong performance on detection tasks Ojha et al. (2023), we select two classical pre-trained models, ResNet50 and CLIP-ViT-L-14 (abbreviated as CLIP), for the empirical study.

### 3.1 QUALITATIVE ANALYSIS

To explore and understand the capability of features in distinguishing real from fake images, we applied t-distributed Stochastic Neighbor Embedding (T-SNE) van der Maaten & Hinton (2008) to the high-level and low-level features extracted by the CNN and ViT architectures, as well as features obtained using our TOFE method.

In Figure 7, the blue and orange points represent feature points of real images and fake images respectively. We can find that the features extracted by ResNet, no matter whether high-level or low-level, can hardly distinguish real images from fake ones across various diffusion types (*e.g.*, PNDM Liu et al. (2022) and DDPM Ho et al. (2020)), as evidenced by the significant overlap of blue and orange sample points. Furthermore, the features extracted by CLIP are insufficient to distinguish between real and fake images (*e.g.*, ADM Dhariwal & Nichol (2021) and IDDPM Nichol & Dhariwal (2021)). In contrast, the representations obtained using our TOFE method clearly distinguish between real and fake distributions.

### 3.2 QUANTITATIVE ANALYSIS

We quantitatively evaluated features extracted by ResNet, CLIP, and TOFE. In Table 1, features processed with T-SNE are assessed using Maximum Mean Discrepancy (MMD) Gretton et al. (2012) and Jensen–Shannon divergence (JS) Lin (1991) to measure the distance between real and fake distributions (*i.e.*, orange vs. blue points in Figure 7). Larger values indicate greater separability and better feature discrimination. For the "ResNet" and "CLIP" rows, we **bold** the higher value between low-level and high-level features. Regarding the "ResNet" row, we can find that the low-level feature is a bit better at distinguishing real and fake images. Regarding the "CLIP" row, the high-level feature is significantly better than the low-level feature but not in all types of diffusion. These results indicate

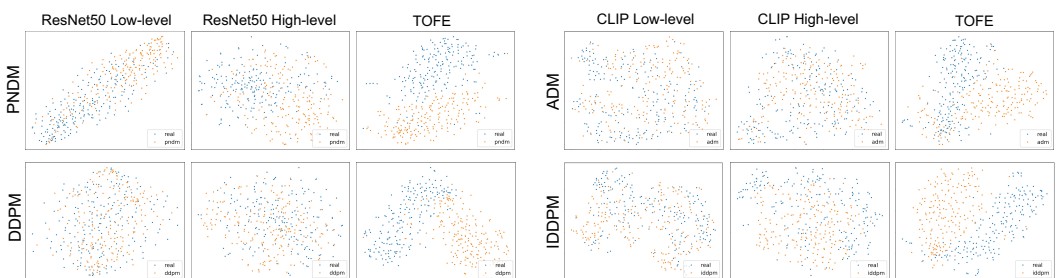

Figure 1: T-SNE visualization of features extracted from ResNet, CLIP, and TOFE.

Table 1: Quantitative results of feature extraction methods across different diffusion types.

| | Evaluation | | ADM | DALLE2 | DDPM | IDDPM | IF | LDM | PNDM | SD-V1 | SD-V2 | VQ-Diffusion | Average |
|---|---|---|---|---|---|---|---|---|---|---|---|---|---|
| **ResNet** | Low-level | MMD↑ | **0.021** | **1.525** | 0.043 | **0.355** | **2.306** | **1.465** | 0.235 | 0.135 | 0.117 | **0.310** | **0.651** |
| | | JS↑ | $9.479 \times 10^{-5}$ | $1.102 \times 10^{-2}$ | $2.126 \times 10^{-4}$ | $8.388 \times 10^{-3}$ | $2.713 \times 10^{-2}$ | $1.106 \times 10^{-2}$ | $1.130 \times 10^{-4}$ | $1.259 \times 10^{-3}$ | $2.689 \times 10^{-4}$ | $5.262 \times 10^{-4}$ | $6.006 \times 10^{-3}$ |
| | High-level | MMD↑ | 0.006 | 0.520 | **0.066** | 0.115 | 1.144 | 0.044 | 0.887 | **1.144** | 0.199 | **1.048** | 0.308 | 0.434 |
| | | JS↑ | $2.275 \times 10^{-5}$ | $8.925 \times 10^{-5}$ | $2.948 \times 10^{-5}$ | $1.024 \times 10^{-5}$ | $8.482 \times 10^{-3}$ | $5.092 \times 10^{-5}$ | $7.498 \times 10^{-3}$ | $1.284 \times 10^{-6}$ | $5.671 \times 10^{-3}$ | $7.867 \times 10^{-4}$ | $1.422 \times 10^{-3}$ |
| | Low & High | MMD↑ | 0.035 | 0.685 | 0.251 | 0.126 | 1.788 | 1.018 | 0.027 | 3.771 | 2.163 | 2.123 | 1.199 |
| | | JS↑ | $1.066 \times 10^{-5}$ | $1.017 \times 10^{-3}$ | $1.212 \times 10^{-3}$ | $7.265 \times 10^{-4}$ | $8.513 \times 10^{-3}$ | $4.203 \times 10^{-4}$ | $9.867 \times 10^{-5}$ | $1.826 \times 10^{-2}$ | $6.513 \times 10^{-3}$ | $1.032 \times 10^{-2}$ | $4.709 \times 10^{-3}$ |
| **CLIP** | Low-level | MMD↑ | **0.104** | 0.749 | **0.227** | 0.028 | 0.997 | **0.353** | 0.565 | 0.153 | 0.318 | 0.908 | 0.440 |
| | | JS↑ | $2.432 \times 10^{-6}$ | $1.055 \times 10^{-5}$ | $4.584 \times 10^{-6}$ | $2.193 \times 10^{-6}$ | $2.165 \times 10^{-5}$ | $5.747 \times 10^{-6}$ | $1.705 \times 10^{-5}$ | $6.475 \times 10^{-6}$ | $1.363 \times 10^{-5}$ | $1.145 \times 10^{-5}$ | $9.578 \times 10^{-6}$ |
| | High-level | MMD↑ | 0.061 | **0.836** | 0.057 | **0.103** | **3.436** | 0.159 | **2.022** | **3.682** | **0.697** | **3.649** | **1.470** |
| | | JS↑ | $6.576 \times 10^{-5}$ | $6.427 \times 10^{-4}$ | $1.119 \times 10^{-4}$ | $3.299 \times 10^{-4}$ | $1.806 \times 10^{-2}$ | $1.946 \times 10^{-4}$ | $9.329 \times 10^{-3}$ | $2.309 \times 10^{-2}$ | $2.615 \times 10^{-3}$ | $2.508 \times 10^{-2}$ | $7.952 \times 10^{-3}$ |
| | Low & High | MMD↑ | 0.019 | 5.145 | 0.931 | 0.456 | 2.105 | 2.245 | 1.398 | 4.463 | 5.753 | 1.274 | 2.379 |
| | | JS↑ | $3.801 \times 10^{-6}$ | $3.409 \times 10^{-2}$ | $3.878 \times 10^{-3}$ | $1.841 \times 10^{-3}$ | $1.001 \times 10^{-2}$ | $7.706 \times 10^{-3}$ | $5.464 \times 10^{-4}$ | $2.969 \times 10^{-2}$ | $4.328 \times 10^{-2}$ | $4.339 \times 10^{-3}$ | $1.354 \times 10^{-2}$ |
| **TOFE (ours)** | | MMD↑ | 1.569 | 5.373 | 2.931 | 2.729 | 3.155 | 5.825 | 2.996 | 5.123 | 5.078 | 2.493 | 3.727 |
| | | JS↑ | $1.159 \times 10^{-2}$ | $1.167 \times 10^{-1}$ | $4.011 \times 10^{-2}$ | $2.464 \times 10^{-2}$ | $1.393 \times 10^{-2}$ | $1.260 \times 10^{-1}$ | $2.459 \times 10^{-2}$ | $1.100 \times 10^{-1}$ | $9.244 \times 10^{-2}$ | $2.521 \times 10^{-2}$ | $5.852 \times 10^{-2}$ |

*__Bold__ shows the higher value between low-level and high-level features.
*Underline marks fused features that exceed both low-level and high-level values.
* Gray cells highlights values of TOFE that exceed all other feature extraction methods.

that low-level and high-level features are not absolutely better than each other; it is not comprehensive enough to build a DeepFake detection method just relying on one of them.

A naive idea is to directly concatenate the high-level and low-level features as the representation for detection. In Table 1, we also calculate the quantitative results of fused low-level and high-level features in the "Low & High" rows. For the "ResNet" and "CLIP" rows, we respectively underline the values that are higher than those in the low- and high-level features. The results show that the fused feature performs better in some cases, indicating that a representation capturing both high-level and low-level features has the potential to benefit the task, although simple concatenation is not sufficient. Thus, we not only need to fuse features, but the fused representation must also effectively support the detection task, which remains a challenge.

In the last row, we show the quantitative results of the features extracted by our TOFE method. We use gray cells to label the values that are higher than all values in the same column (*i.e.*, same diffusion type). The results show that the features extracted by TOFE show clear advantages in distinguishing real and fake images, facilitating more effective learning for the classifier.

## 4 TEXT MODALITY ORIENTED FEATURE EXTRACTION

Inspired by the understanding that the text of the text-to-image (T2I) model is a high-level semantic representation, our TOFE method aims to obtain the representation of the target image by refining the embedding of such text to contain fine-grained details (*i.e.*, low-level features). Note that although the text of the text-to-image model is a high-level semantic representation, this is because it is made up of discrete tokens. The embedding of the text is in continuous space and can be moved continuously in the embedding space (with optimization) to a target embedding representation that contains high-level and low-level information.

### 4.1 PRELIMINARIES

Our feature extraction method is over the Latent Diffusion Model (LDM) Rombach et al. (2022), a variant of the Denoising Diffusion Probabilistic Model (DDPM) Ho et al. (2020) that operates in the latent space of an autoencoder.

**Latent diffusion model.** The conditional text-to-image LDM is designed to map a noise vector $\mathbf{z}_T$ and text condition $\mathcal{Q}$ to an output latent vector $\mathbf{z}_0$. In order to perform sequential denoising, the network $\epsilon_\theta(\cdot)$ is trained to predict the noise at each timestep, following the objective:

$$\min_\theta \mathbb{E}_{\mathbf{z},\epsilon \sim \mathcal{N}(0,1), t \sim \text{Uniform}(1,T)} \|\epsilon - \epsilon_\theta(\mathbf{z}_t, t, \mathcal{C})\|_2^2, \tag{1}$$

where $\mathcal{C} = \Psi(\mathcal{Q})$ is the embedding of the text condition $\mathcal{Q}$, symbol $T$ is the total time steps in the sequential denoising procedure, and $\mathbf{z}_t$ is a noise sample at timestep $t$. When denoising the latent, given the noise vector $\mathbf{z}_T$, the noises sequentially predicted by $\epsilon_\theta(\cdot)$ can be gradually removed for $T$ steps with DDIM sampling Song et al. (2021),

$$\mathbf{z}_{t-1} = \sqrt{\frac{\alpha_{t-1}}{\alpha_t}} \mathbf{z}_t + \sqrt{\alpha_{t-1}} \left( \sqrt{\frac{1}{\alpha_{t-1}} - 1} - \sqrt{\frac{1}{\alpha_t} - 1} \right) \cdot \epsilon_\theta(\mathbf{z}_t), \tag{2}$$

where the definition of $\alpha_t$ can refer to DDIM.

**Classifier-free guidance.** For conditional text-to-image generation, Ho et al. Ho & Salimans (2021) proposes the classifier-free guidance technique, which fuses the predictions performed conditionally and unconditionally to guide the sampling procedure, which can generate arbitrary image categories. To be specific, let $\oslash = \Psi("")$ to be the embedding of a null text; the prediction is defined by

$$\hat{\epsilon}_\theta(\mathbf{z}_t, t, \mathcal{C}, \oslash) = w * \epsilon_\theta(\mathbf{z}_t, t, \mathcal{C}) + (1 - w) * \epsilon_\theta(\mathbf{z}_t, t, \oslash), \tag{3}$$

where $w$ is the guidance scale parameter and $w = 7.5$ is the default in Stable Diffusion Rombach et al. (2022) (*i.e.*, a popular LDM variant).

**DDIM inversion.** In contrast to DDIM sampling, DDIM Song et al. (2021) also propose a simple inversion technique that can gradually add noise $\mathbf{z}_0$ for $T$ timesteps to achieve $\mathbf{z}_T$ (see Figure 2(a)). The method is based on the assumption that the ordinary differential equation (ODE) process can be reversed in limited small steps that:

$$\mathbf{z}_{t+1} = \sqrt{\frac{\alpha_{t+1}}{\alpha_t}} \mathbf{z}_t + \sqrt{\alpha_{t+1}} \left( \sqrt{\frac{1}{\alpha_{t+1}} - 1} - \sqrt{\frac{1}{\alpha_t} - 1} \right) \cdot \epsilon_\theta(\mathbf{z}_t). \tag{4}$$

### 4.2 PROBLEM FORMULATION AND SOLUTION

#### 4.2.1 OBSERVATION AND MOTIVATION

As shown in Figure 2(a), given a vector $\mathbf{z}_0$ which is latent of an image $I$, we can use DDIM inversion to reverse $T$ timesteps to achieve the latent $\mathbf{z}_T$ (in this example, $T = 50$). With the $\mathbf{z}_T$, as shown in Figure 2(b), using classifier-free guidance to generate images with different text conditions $\mathcal{Q}$ will lead to distinct output latents which represent various image contents. There are two observations. ❶ Regarding embedding of the first (orange) text condition, due to its completely inconsistent high-level infor-

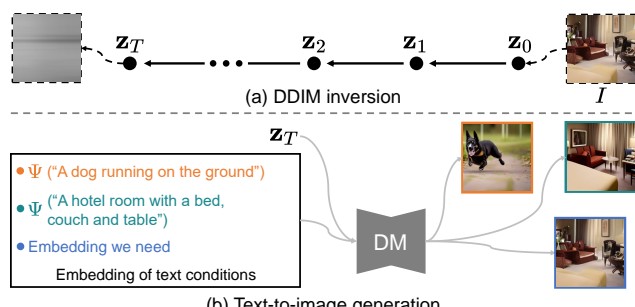

(a) DDIM inversion

(b) Text-to-image generation

Figure 2: T2I generation with different conditions.

mation from the image $I$, the generated image (image with orange border) is significantly different from image $I$. ❷ Regarding embedding of second (teal) text condition, it is extracted from $I$ by BLIP Li et al. (2022), which shares consistent high-level semantic information with image $I$. However, due to a lack of description of fine-grained details in the embedding, relying solely on high-level semantic information can only result in an image (image with teal border) that differs greatly in details from the image $I$. From the previous observations, we can find that, **in order to reconstruct the image $I$, we need an embedding (blue) that contains high-level and low-level features together and it is the representation that we need for further detection tasks.** The problem is how to obtain such embedding representation.

### 4.2.2 PROBLEM DEFINITION

Given a target real or fake image $I$ (corresponding latent is $\mathbf{z}_0$) and a pre-trained conditional text-to-image LDM $DM(\cdot)$, the latent trajectory $\mathbf{z}_1, \ldots, \mathbf{z}_T$ is achieved by DDIM inversion operation with $t = 1, \ldots, T$ timesteps respectively. In order to reconstruct $\mathbf{z}_0$, the procedure should start with latent $\mathbf{z}_T$ and perform classifier-free guidance generation with the same condition $\mathcal{C}_t$ (embedding of a token text with high-level semantic) at each timestep $t$ to follow the reverse latent trajectory (*i.e.*, $\mathbf{z}_T, \mathbf{z}_{T-1}, \ldots, \mathbf{z}_0$). For each timestep $t$, due to the coarse-grained description of $\mathcal{C}_t$, there is a deviation between the generated latent $\mathbf{z}_{t-1}^*$ and the ground truth latent $\mathbf{z}_{t-1}$ from trajectory. Our goal is to obtain $\hat{\mathcal{C}}_t$ that can make the $\mathbf{z}_{t-1}^*$ same as $\mathbf{z}_{t-1}$ by refining the condition $\mathcal{C}_t$. There $\hat{\mathcal{C}}_t$ is a representation that can guide the generation of the target image, which means it captures the high-level and low-level information of the target image (satisfying our requirement).

### 4.2.3 OBJECTIVE

The idea is to start from a condition embedding $\mathcal{C}$ (*i.e.*, $\mathcal{C} = \Psi(\mathcal{Q})$) of a token text condition $\mathcal{Q}$ (*e.g.*, $\mathcal{Q}$="a dog") and iteratively optimize it to be $\hat{\mathcal{C}}$. Specifically, as shown in Figure 3, at timestep $t$, given latent $\mathbf{z}_t$, using DDIM sampling (Eq. (2)), classifier-free guidance (Eq. (3)) and condition $\mathcal{C}_t$, we can achieve $\mathbf{z}_{t-1}^*$. However, due to the imprecise description of condition $\mathcal{C}_t$, the latent $\mathbf{z}_{t-1}^*$ will a bit deviate from the trajectory (*i.e.*, $\mathbf{z}_t, \mathbf{z}_{t-1}, \ldots, \mathbf{z}_0$) for generating target latent $\mathbf{z}_0$. That is, we need to seek a precise condition $\hat{\mathcal{C}}_t$ that can guide the generation of the latent $\mathbf{z}_{t-1}$ ($\mathbf{z}_{t-1}$ is obtained with DDIM inversion from $\mathbf{z}_0$). By calculating the mean square error (MSE) between $\mathbf{z}_{t-1}$ and $\mathbf{z}_{t-1}^*$ as loss, the objective is

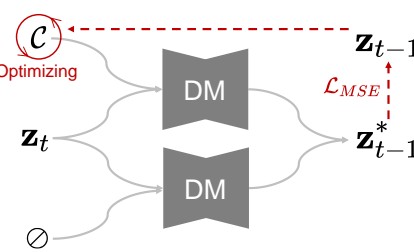

Figure 3: Optimization of conditional embeddings.

$$\hat{\mathcal{C}} = \arg\min_{\mathcal{C}} \|\mathbf{z}_{t-1} - \mathbf{z}_{t-1}^*\|_2^2. \tag{5}$$

Then, for the full timesteps $t = T, \ldots, 1$, we optimize $N$ iterations for each condition $\mathcal{C}_t$ and totally achieve $T$ condition embeddings. Each optimized condition embedding $\hat{\mathcal{C}}_t$ captures high-level and low-level features, which is the representation we need for further detection tasks. With the extracted features, we can easily construct a detection model by training a simple classifier (*i.e.*, MLP). Note that although the proposed TOFE can obtain representation capturing both high-level and low-level features while is effective for the DeepFake detection task. It does not imply that TOFE is the best text modality-oriented feature extraction method and this does not mean any representation that contains high-level & low-level features can benefit the detection task. Additionally, while TOFE performs well for the detection task, this does not preclude the feasibility of other methods, possibly image modality-oriented feature extraction methods.

### 4.3 ALGORITHM

The algorithm of the TOFE method is intuitive and is shown in Algorithm 1. As an optimization algorithm, in addition to using the condition embedding and latent of the image as inputs, there are two additional parameters: learning rate $\eta$ and iteration number $N$. In line 1, the algorithm first uses DDIM inversion to obtain the latent sequence in Figure 2(a). In lines 2 to 4, for

---

**Algorithm 1:** Text modality Oriented Feature Extraction

**Input:** Condition embeddings $\{\mathcal{C}_t\}_{t=1}^T$, Latent $\mathbf{z}_0$ from $I$, Learning rate $\eta$, Iteration number $N$
**Output:** Optimized condition embeddings $\{\hat{\mathcal{C}}_t\}_{t=1}^T$
1 $\{\mathbf{z}_t\}_{t=0}^T \leftarrow$ **Inversion**$(\mathbf{z}_0)$ ▷ Compute $\mathbf{z}_0$ via DDIM inversion
2 **for** $t = T, T-1, \ldots, 1$ **do**
3      **for** $i = 0, 1, \ldots, N-1$ **do**
4          $\mathcal{C}_t \leftarrow \mathcal{C}_t - \eta \nabla_{\mathcal{C}} \|\mathbf{z}_{t-1} - \mathbf{z}_{t-1}^*\|_2^2$
5 **return** $\{\hat{\mathcal{C}}_t\}_{t=1}^T$

---

each pair of latent $\mathbf{z}_{t-1}$ and condition embedding $\mathcal{C}_t$ in latent sequence, the algorithm iteratively optimize the condition embedding according to the difference between latent $\mathbf{z}_{t-1}$ and target latent $\mathbf{z}_{t-1}^*$. At last, in line 5, the algorithm returns the refined condition embedding list.

Table 2: Detection performance comparison with representative detection methods.

| ACC(%)/AP(%) | Diffusion Types | | | | | | | | | | Average |
|---|---|---|---|---|---|---|---|---|---|---|---|
| | ADM* | PNDM* | IDDPM* | DDPM | IF | LDM | DALLE2 | SD-V1 | SD-V2 | VQ-Diffusion | |
| CNNDet Wang et al. (2020) | 96.95/99.69 | 99.50/99.98 | 99.30/99.97 | 98.34/99.56 | 86.15/96.27 | 91.10/98.48 | 86.66/92.93 | 81.35/95.69 | 77.20/93.26 | 83.75/97.08 | 90.03/97.29 |
| UFD Ojha et al. (2023) | 90.30/97.81 | 98.30/99.95 | 98.15/99.87 | 98.50/99.90 | 54.50/74.32 | 65.95/84.98 | 95.10/99.01 | 83.15/95.74 | 79.80/93.74 | 98.25/99.89 | 86.20/94.52 |
| DIRE Wang et al. (2023) | 87.60/99.59 | 94.60/99.87 | 90.35/99.67 | 92.42/99.54 | 98.03/98.92 | 87.95/99.51 | 98.15/99.91 | 96.60/99.90 | 98.60/99.97 | 91.65/99.79 | 93.60/99.67 |
| DMDet Corvi et al. (2023) | 99.40/99.98 | 98.95/99.98 | 99.95/99.98 | 99.95/99.99 | 67.31/98.21 | 93.93/99.90 | 53.65/97.53 | 66.95/99.49 | 53.56/90.92 | 97.90/99.98 | 78.49/98.60 |
| AEROBLADE Ricker et al. (2024) | -/74.26 | -/49.77 | -/60.10 | -/53.66 | -/97.02 | -/99.72 | -/67.48 | -/79.63 | -/87.94 | -/88.88 | -/75.85 |
| TOFE (ours) | 97.35/99.73 | 97.65/99.39 | 98.35/99.60 | 99.05/99.92 | 98.55/99.87 | 98.20/99.81 | 99.15/99.95 | 97.60/99.76 | 96.90/99.59 | 99.30/99.98 | 98.21/99.76 |

*Cyan cells denote where TOFE achieves the highest ACC among all detection methods.

# 5 EXPERIMENT

## 5.1 EXPERIMENTAL SETUPS

**Datasets and models.** In evaluations, we use the DIRE dataset. It contains ten diffusion types, *i.e.*, ADM Dhariwal & Nichol (2021), DDPM Ho et al. (2020), IDDPM Nichol & Dhariwal (2021), DALLE2 Ramesh et al. (2022), IF deep floyd (2023), PNDM Liu et al. (2022), VQ-Diffusion Gu et al. (2022), LDM Rombach et al. (2022), Stable Diffusion v1 and v2. Each type has thousands of fake images. The images are all with size $256 \times 256$. The baselines are CNNDet Wang et al. (2020), DIRE Wang et al. (2023), UFD Ojha et al. (2023), DMDet Corvi et al. (2023), AEROBLADE Ricker et al. (2024). They are all published representative detection works for diffusion-based DeepFake.

**Metrics.** Following the metrics in baselines Wang et al. (2020; 2023); Ojha et al. (2023); Corvi et al. (2023); Ricker et al. (2024), we use average accuracy (ACC) and average precision (AP) as the metrics for evaluating the detection performance. The threshold for computing accuracy is set to 0.5 following Wang et al. (2023). Besides, the f1-score, recall, and area under the curve value (AUC) metrics are provided in the Appendix.

**Implementation details of our method.** The LDM we used is Stable Diffusion v1.4. For the learning rate $\eta$ and iteration number $N$, the value is 0.01 and 10. The default condition embedding $\mathcal{C}$ is $\oslash = \Psi("")$. Using $\oslash$ is general for any image and resource-consuming (does not need to generate a caption for each image). The timestep $T$ is 1 and the reason is that the time consumption of feature extraction is faster than that of other values and the extracted feature is good for detecting real and fake images. Note that when $T$ is other values (*e.g.*, 50), the features are also good enough, as shown in Section A.4.3. The interconversion of images and corresponding latents are based on the autoencoder of the pre-trained stable diffusion model. The self-built classifier for feature analysis is a simple MLP that only contains four layers and cross-entropy loss. The learning rate for the classifier is set to $1 \times 10^{-5}$ and the classifier is trained for 10,000 iterations. All the experiments were run on an Ubuntu system with two NVIDIA A6000 Tensor Core GPUs of 48G RAM.

## 5.2 COMPARISON WITH DETECTION BASELINES

As a DeepFake detection method, we compare our TOFE method with five representative detection methods, *i.e.*, CNNDet Wang et al. (2020), UFD Ojha et al. (2023), DIRE Wang et al. (2023), DMDet Corvi et al. (2023), and AEROBLADE Ricker et al. (2024). To verify the in-domain (ID) and out-of-domain (OOD) performance of our method, we follow the common practices "train-on-many and test-on-many" in DeepFake detection Wang et al. (2023), that is, train on some diffusion types and test on more. To be specific, following DIRE Wang et al. (2023), we train our classifier on 30,000 images generated by three types of diffusion models (*i.e.*, ADM, PNDM, and IDDPM, with * in Table) and 30,000 real images, a total of 60,000 images. For the test dataset, there are 10 different diffusion types (*i.e.*, ADM, DALLE2, DDPM, IDDPM, IF, LDM, PNDM, SD-V1, SD-V2, and VQ-Diffusion). For each diffusion type, there are 1,000 real images and 1,000 fake images, a total of 2,000 images per type. The baselines (CNNDet, UFD, DIRE, DMDet, and AEROBLADE) are all trained and tested on the above dataset with their own experimental setting. Since AEROBLADE does not provide a specific way to calculate ACC based on their special setting, we just provide the AP value as they do.

As shown in Table 2, we report ACC (%) and AP (%) (ACC/AP in the Table). We can find that, although the baselines all show good performance on the ten diffusion types, our TOFE demonstrates significant performance advantages in five diffusion types (*i.e.*, ADM, DDPM, IF, DALLE2, and VQ-Diffusion) with the detection results achieving the best ACC and AP. Specifically, for in-domain testing (*i.e.*, results on ADM, PNDM, and IDDPM), CNNDet, UFD, DIRE, and our method TOFE

Table 3: Comparison with classical feature extraction methods (all with same classifier architecture).

| ACC(%)/AP(%) | | ADM* | PNDM* | IDDPM* | DDPM | Diffusion Types IF | LDM | DALLE2 | SD-V1 | SD-V2 | VQ-Diffusion | Average |
|---|---|---|---|---|---|---|---|---|---|---|---|---|
| **ResNet** | Low-level | 72.45/79.26 | 72.45/76.58 | 76.95/82.04 | 73.20/64.56 | 62.60/65.05 | 70.65/73.82 | 54.35/38.71 | 59.35/61.03 | 65.20/69.38 | 68.90/73.42 | 67.61/68.38 |
| | High-level | 92.15/97.83 | 96.10/99.44 | 96.15/99.46 | 94.80/97.90 | 93.80/84.30 | 84.40/94.06 | 48.45/27.99 | 47.30/41.00 | 47.45/38.69 | 60.25/76.70 | 76.09/75.74 |
| | Low & High | 94.09/97.99 | 98.88/99.47 | 98.78/99.48 | 97.51/98.14 | 64.01/84.53 | 85.63/94.88 | 15.88/28.17 | 20.33/39.96 | 47.95/94.20 | 55.70/79.80 | 67.88/81.66 |
| **CLIP** | Low-level | 81.95/91.31 | 91.15/97.87 | 87.20/95.33 | 89.75/93.90 | 70.95/81.71 | 84.80/92.79 | 85.05/87.05 | 77.15/86.16 | 77.80/86.89 | 90.15/96.84 | 83.60/90.98 |
| | High-level | 97.85/99.81 | 99.70/99.99 | 99.45/99.98 | 99.15/99.93 | 87.80/98.61 | 83.10/97.97 | 94.80/99.31 | 94.90/99.66 | 94.50/99.60 | 99.45/99.98 | 95.07/99.49 |
| | Low & High | 97.80/99.85 | 99.80/100.00 | 99.60/99.99 | 99.35/99.94 | 87.40/98.80 | 92.15/99.52 | 93.80/99.23 | 95.85/99.75 | 97.05/99.80 | 99.70/100.00 | 96.25/99.69 |
| **TOFE (ours)** | | 97.35/99.73 | 97.65/99.39 | 98.35/99.60 | 99.05/99.92 | 98.55/99.87 | 98.20/99.81 | 99.15/99.95 | 97.60/99.76 | 96.90/99.59 | 99.30/99.98 | **98.21/99.76** |

*Cyan cells denote where TOFE achieves the highest ACC.

all show good performance. For out-of-domain testing, our method and DIRE achieve the best performance, *i.e.*, ACC higher than 90% and AP higher than 99% in most cases, which shows strong generalization ability. To summarize, as shown in the last column, **compared with five representative baselines, our method achieves the best average performance** (*i.e.*, 98.21(%) ACC and 99.76(%) AP). In addition, we also present other metrics (*e.g.*, F1-Score, Recall, and AUC) to provide a comprehensive performance evaluation in Appendix (Table 11, Table 12, and Table 13). TOFE demonstrates **the best average performance across all other metrics**.

## 5.3 COMPARISON WITH OTHER FEATURE EXTRACTION METHODS

Although the features extracted by our TOFE method make it easier to distinguish real and fake images than other feature extraction methods, this does not mean that the detection performance of our method is absolutely better since the detection results depend on what the classifier learns. Therefore, we employ the same simple classifier (MLP with only four layers) to process the features from different methods and the detection results are shown in Table 3.

We report ACC (%) and AP (%) (ACC/AP in the Table). we can find that the detection results of our method achieve **the best ACC and AP on average**. From the table, we can also find two interesting observations. ❶ In "ResNet" row, the results of "Low & High" is partially better than that of "Low-level" and "High-level" while in "CLIP" row, "Low & High" is absolutely better than that of "Low-level" and "High-level" on both ACC and AP. This means the effect of simply concatenating features from high-level and low-level for the DeepFake detection task is unstable, which highlights the necessity of finding a good fusion of the features. ❷ The detection performance of "Low-level", "High-level" and "Low & High" in "CLIP" row is always better than that in "ResNet" row. This provides evidence of the advantages of features extracted from CLIP (foundation model trained with a large dataset), which is consistent with the conclusion in Ojha et al. (2023).

## 5.4 TRANSFERABILITY EVALUATION ON GAN-BASED DEEPFAKE

In the image synthesis task, GAN is another mainstream approach for DeepFake generation. Thus we further explore the transferability of our detection method on the GAN-based DeepFake. Here we choose the classical and well-known types such as ProGAN Karras et al. (2018), StarGAN Choi et al. (2018), and StyleGAN Karras et al. (2019). For testing on each type, there are 1,000 real and 1,000 fake images. The detection models of CNNDet, UFD, DIRE, DMDet, AER-OBLADE, and our TOFE are the same as in Table 2. In Table 4, our method achieves the best ACC and AP, which are both higher than 95%.

Table 4: Transferability on GAN-based DeepFake.

| ACC(%)/AP(%) | GAN Types ProGAN | StyleGAN | StarGAN | Average |
|---|---|---|---|---|
| **CNNDet** Wang et al. (2020) | 56.14/82.01 | 51.10/70.93 | 56.30/73.09 | 54.51/75.34 |
| **UFD** Ojha et al. (2023) | 97.80/99.74 | 96.70/99.51 | 94.16/83.90 | 96.22/94.38 |
| **DIRE** Wang et al. (2023) | 86.00/99.19 | 91.95/99.74 | 83.00/91.48 | 86.98/96.80 |
| **DMDet** Corvi et al. (2023) | 89.16/90.42 | 95.58/93.76 | 90.65/96.73 | 91.80/93.64 |
| **AEROBLADE** Ricker et al. (2024) | -/49.33 | -/48.82 | -/43.10 | -/47.08 |
| **TOFE (ours)** | 97.30/99.66 | 97.95/99.78 | 95.30/98.21 | **96.85/99.22** |

*Cyan cells denote the TOFE achieves the highest ACC.

## 5.5 RECONSTRUCTION QUALITY

We aim to develop a representation that guides the reconstruction of a target image with a pre-trained T2I model. Specifically, we posit that if the representation successfully reconstructs the target image, it must capture both high-level and low-level features, thereby serving as strong evidence of its comprehensiveness. To assess whether the representation meets these requirements, we evaluate the reconstruction quality. Compared with the target image, we use metrics Peak Signal-to-Noise Ratio

Table 5: Quality of reconstructed images.

| | Real | ADM | DALLE2 | DDPM | IDDPM | IF | LDM | PNDM | SD-V1 | SD-V2 | VQ-Diffusion |
|---|---|---|---|---|---|---|---|---|---|---|---|
| **PSNR ↑** | 31.486 | 35.391 | 33.808 | 35.618 | 35.001 | 38.754 | 41.485 | 31.678 | 30.025 | 32.203 | 36.190 |
| **SSIM ↑** | 0.891 | 0.935 | 0.911 | 0.942 | 0.927 | 0.970 | 0.977 | 0.871 | 0.874 | 0.873 | 0.943 |

(PSNR) Hore & Ziou (2010) and Structural Similarity (SSIM) Wang et al. (2004) to calculate the quality of the reconstructed image (average of 200 images per type). As shown in Table 5, we find the PSNR values are all above 30, which means that the reconstructed image and target image are visually identical images Huang & Sakurai (2011). For SSIM, most of the values are larger or near 0.9, which also verifies the high quality of the reconstructed image.

## 5.6 ABLATION STUDY

### 5.6.1 LEARNING RATE

The choices of the learning rate are from $1 \times 10^{-2}$ to $1 \times 10^{-6}$. In Table 6, following the setting of Table 1, we demonstrate the MMD and JS values under different learning rates. We can find that when the learning rate $\eta$ is $1 \times 10^{-2}$, the MMD and JS value of distributions between real and fake distributions in T-SNE is the highest. Also, we can find that although not the best, the MMD and JS values under $1 \times 10^{-3}$ to $1 \times 10^{-5}$ are comparable or better than that of ResNet and CLIP in Table 1.

Table 6: Ablation on learning rate $\eta$.

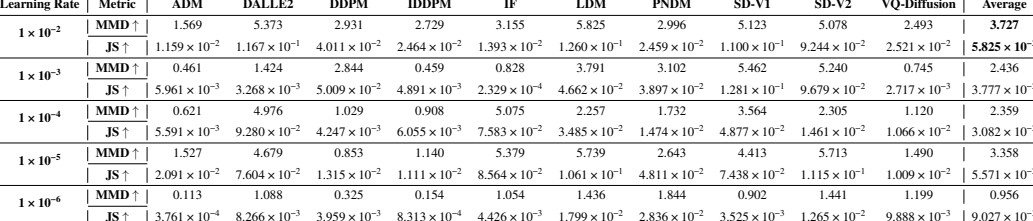

| Learning Rate | Metric | ADM | DALLE2 | DDPM | IDDPM | IF | LDM | PNDM | SD-V1 | SD-V2 | VQ-Diffusion | Average |
|---|---|---|---|---|---|---|---|---|---|---|---|---|
| $1 \times 10^{-2}$ | MMD↑ | 1.569 | 5.373 | 2.931 | 2.729 | 3.155 | 5.825 | 2.996 | 5.123 | 5.078 | 2.493 | **3.727** |
| | JS↑ | $1.159 \times 10^{-2}$ | $1.167 \times 10^{-1}$ | $4.011 \times 10^{-2}$ | $2.464 \times 10^{-2}$ | $1.393 \times 10^{-2}$ | $1.260 \times 10^{-1}$ | $2.459 \times 10^{-2}$ | $1.100 \times 10^{-1}$ | $9.244 \times 10^{-2}$ | $2.521 \times 10^{-2}$ | **$5.825 \times 10^{-2}$** |
| $1 \times 10^{-3}$ | MMD↑ | 0.461 | 1.424 | 2.844 | 0.459 | 0.828 | 3.791 | 3.102 | 5.462 | 5.240 | 0.745 | 2.436 |
| | JS↑ | $5.961 \times 10^{-3}$ | $3.268 \times 10^{-3}$ | $5.009 \times 10^{-2}$ | $4.891 \times 10^{-3}$ | $2.329 \times 10^{-4}$ | $4.662 \times 10^{-2}$ | $3.897 \times 10^{-2}$ | $1.281 \times 10^{-1}$ | $9.679 \times 10^{-2}$ | $2.717 \times 10^{-3}$ | $3.777 \times 10^{-2}$ |
| $1 \times 10^{-4}$ | MMD↑ | 0.621 | 4.976 | 1.029 | 0.908 | 5.075 | 2.257 | 1.732 | 3.564 | 2.305 | 1.120 | 2.359 |
| | JS↑ | $5.591 \times 10^{-3}$ | $9.280 \times 10^{-2}$ | $4.247 \times 10^{-3}$ | $6.055 \times 10^{-3}$ | $7.583 \times 10^{-2}$ | $3.485 \times 10^{-2}$ | $1.474 \times 10^{-2}$ | $4.877 \times 10^{-2}$ | $1.461 \times 10^{-2}$ | $1.066 \times 10^{-2}$ | $3.082 \times 10^{-2}$ |
| $1 \times 10^{-5}$ | MMD↑ | 1.527 | 4.679 | 0.853 | 1.140 | 5.379 | 5.739 | 2.643 | 4.413 | 5.713 | 1.490 | 3.358 |
| | JS↑ | $2.091 \times 10^{-2}$ | $7.604 \times 10^{-2}$ | $1.315 \times 10^{-2}$ | $1.111 \times 10^{-2}$ | $8.564 \times 10^{-2}$ | $1.061 \times 10^{-1}$ | $4.811 \times 10^{-2}$ | $7.438 \times 10^{-2}$ | $1.115 \times 10^{-1}$ | $1.009 \times 10^{-2}$ | $5.571 \times 10^{-2}$ |
| $1 \times 10^{-6}$ | MMD↑ | 0.113 | 1.088 | 0.325 | 0.154 | 1.054 | 1.436 | 1.844 | 0.902 | 1.441 | 1.199 | 0.956 |
| | JS↑ | $3.761 \times 10^{-4}$ | $8.266 \times 10^{-3}$ | $3.959 \times 10^{-3}$ | $8.313 \times 10^{-4}$ | $4.426 \times 10^{-3}$ | $1.799 \times 10^{-2}$ | $2.836 \times 10^{-2}$ | $3.525 \times 10^{-3}$ | $1.265 \times 10^{-2}$ | $9.888 \times 10^{-3}$ | $9.027 \times 10^{-3}$ |

### 5.6.2 PRE-TRAINED T2I MODEL VERSIONS AND TIMESTEPS

We investigate the effects of the pre-trained T2I model version and the number of timesteps on TOFE's detection performance. Table 7 shows that the detection performance of TOFE remains consistently high across all diffusion models, with only minor differences observed when changing the Stable Diffusion version (v1.4 vs. v2.0) or increasing the timestep to $T = 50$.

Table 7: Detection performance for T2I model versions and timesteps $T$.

| ACC(%)/AP(%) | Diffusion Types | | | | | | | | | Average |
|---|---|---|---|---|---|---|---|---|---|---|
| | ADM $^*$ | PNDM $^*$ | IDDPM $^*$ | DDPM | IF | LDM | DALLE2 | SD-V1 | SD-V2 | VQ-Diffusion | |
| **TOFE With SD-V1.4** | 97.35/99.73 | 97.65/99.39 | 98.35/99.60 | 99.05/99.92 | 98.55/99.87 | 98.20/99.81 | 99.15/99.95 | 97.60/99.76 | 96.90/99.59 | 99.30/99.98 | **98.21/99.76** |
| **TOFE With SD-V2.0** | 97.70/99.72 | 98.70/99.90 | 98.75/99.95 | 98.40/99.66 | 97.35/99.73 | 98.25/99.77 | 97.50/99.37 | 96.20/99.50 | 93.45/98.94 | 98.45/99.94 | **97.48/99.65** |
| **TOFE With T = 50** | 96.25/99.61 | 96.50/99.54 | 96.45/99.82 | 96.20/99.22 | 95.75/99.41 | 96.10/99.53 | 95.85/98.89 | 95.80/99.28 | 93.55/98.12 | 96.45/99.85 | **95.89/99.33** |

## 6 CONCLUSION

Detecting DeepFake images generated by diffusion models is increasingly challenging due to their high realism. Feature analysis shows that existing detection methods that rely on high-level image features struggle to capture the complexities of diffusion-based fakes. We introduce a Text Modality-Oriented Feature Extraction method, which captures both high-level and low-level features by translating image features into a text-based domain. This method significantly enhances the ability to distinguish between real and fake images, outperforming existing representative detection techniques across multiple diffusion models. In future work we will explore the potential of text-based feature representations to further advance DeepFake detection.

## ETHICS STATEMENT

This work adheres to the ICLR Code of Ethics. Our study focuses on developing methods to improve the reliability of detecting diffusion-based synthetic media. The methods do not involve human subjects, private data, or activities that raise privacy or safety concerns. While highly realistic synthetic images could be misused, our work is intended solely to enhance detection and mitigate potential misuse. We encourage responsible application of generative technologies and careful consideration of ethical implications in real-world scenarios.

## REPRODUCIBILITY STATEMENT

We are committed to ensuring the reproducibility of results in accordance with ICLR standards. The main text presents feature analysis (Section 3) and the TOFE design with text embedding refinement (Section 4). Section 5 describes datasets, models, metrics, and implementation details. Additional details and extended experiments are provided in the appendix. Upon paper acceptance, we will publicly release the codebase and all scripts necessary to reproduce our experiments and main results.

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

# A    APPENDIX

## A.1    USE OF LARGE LANGUAGE MODELS

Large Language Models (LLMs) were used solely as a general-purpose tool to aid and polish the writing. No substantive research ideation or experimental design was performed by the LLMs.

## A.2    DATASET PROCESSING AND PREPROCESSING

In our experiments on diffusion-based DeepFake detection, we identified a potential bias in the DIRE dataset Wang et al. (2023), where real images are JPEG-compressed while fake images are uncompressed. Such a discrepancy could allow a detector to rely on spurious cues arising from compression artifacts rather than genuine generative traces, as discussed in prior work Ricker et al. (2024). To mitigate this issue, we preprocess both real and fake images to ensure consistent JPEG compression parameters, including compression quality and chroma subsampling, across all samples. All experiments reported in this paper are conducted under this controlled setup, ensuring that the observed detection performance reflects the true efficacy of the proposed TOFE method rather than differences induced by image format or compression.

## A.3    DETAILS OF LOW LEVEL AND HIGH LEVEL IN RESNET AND CLIP

For ResNet50, we extract the output of "layer1" (dimension is [1,256,56,56]) as the low-level feature and extract the output of "average pooling" (prior layer of the linear layer, after "layer4", dimension is [1,2048,1,1]) as the high-level feature.

For CLIP-ViT-L-14, we extract the output of visual transformer "resblocks0" (dimension is [257,1,1024]) as the low-level feature and extract the output of "layernorm" (prior layer of the linear layer, after transformer "resblocks11", dimension is [1,768]) as the high-level feature.

## A.4    ABLATION STUDY

### A.4.1    ITERATION NUMBER

The choices of the iteration number are from 10 to 50. As shown in Table 8, we can find that with different iteration numbers, the average MMD and JS have small differences. Additionally, the values are **all better than** the highest MMD / JS achieved by ResNet ($1.199$ / $6.006 \times 10^{-3}$) and CLIP ($2.379$ / $1.354 \times 10^{-2}$) in Table 1. Considering computational efficiency, we choose $N = 10$ as the setting for TOFE.

### A.4.2    VERSION OF T2I MODEL IN TOFE

We use Stable Diffusion v1.4 as the pre-trained text-to-image model in TOFE for feature extraction in the above experiments. Here we make an ablation study of its version to Stable Diffusion v2.0. For

Table 8: Ablation on iteration number $N$.

| Iteration Number | Metric | ADM | DALLE2 | DDPM | IDDPM | IF | LDM | PNDM | SD-V1 | SD-V2 | VQ-Diffusion | Average |
|---|---|---|---|---|---|---|---|---|---|---|---|---|
| 10 | MMD ↑ | 1.569 | 5.373 | 2.931 | 2.729 | 3.155 | 5.825 | 2.996 | 5.123 | 5.078 | 2.493 | **3.727** |
| | JS ↑ | $1.159 \times 10^{-2}$ | $1.167 \times 10^{-1}$ | $4.011 \times 10^{-2}$ | $2.464 \times 10^{-2}$ | $1.393 \times 10^{-2}$ | $1.260 \times 10^{-1}$ | $2.459 \times 10^{-2}$ | $1.100 \times 10^{-1}$ | $9.244 \times 10^{-2}$ | $2.521 \times 10^{-2}$ | $5.825 \times 10^{-2}$ |
| 20 | MMD ↑ | 0.321 | 5.887 | 2.689 | 0.982 | 2.657 | 6.139 | 1.919 | 4.868 | 5.469 | 1.442 | 3.237 |
| | JS ↑ | $1.833 \times 10^{-3}$ | $1.383 \times 10^{-1}$ | $3.128 \times 10^{-2}$ | $1.322 \times 10^{-2}$ | $4.798 \times 10^{-2}$ | $1.486 \times 10^{-1}$ | $1.182 \times 10^{-2}$ | $8.151 \times 10^{-2}$ | $1.220 \times 10^{-1}$ | $3.333 \times 10^{-3}$ | $5.999 \times 10^{-2}$ |
| 30 | MMD ↑ | 1.910 | 5.750 | 0.510 | 1.809 | 1.679 | 4.948 | 1.130 | 5.529 | 6.011 | 2.091 | 3.137 |
| | JS ↑ | $1.567 \times 10^{-2}$ | $1.334 \times 10^{-1}$ | $6.156 \times 10^{-3}$ | $2.270 \times 10^{-2}$ | $1.105 \times 10^{-2}$ | $9.560 \times 10^{-2}$ | $7.487 \times 10^{-3}$ | $1.187 \times 10^{-1}$ | $1.478 \times 10^{-1}$ | $1.350 \times 10^{-2}$ | $5.720 \times 10^{-2}$ |
| 40 | MMD ↑ | 1.213 | 5.834 | 0.089 | 0.131 | 2.269 | 6.304 | 0.732 | 4.108 | 4.674 | 1.038 | 2.639 |
| | JS↑ | $1.176 \times 10^{-2}$ | $1.386 \times 10^{-1}$ | $1.287 \times 10^{-4}$ | $1.395 \times 10^{-4}$ | $1.100 \times 10^{-2}$ | $1.473 \times 10^{-1}$ | $9.331 \times 10^{-4}$ | $7.888 \times 10^{-2}$ | $7.943 \times 10^{-2}$ | $1.391 \times 10^{-4}$ | $4.684 \times 10^{-2}$ |
| 50 | MMD ↑ | 2.524 | 5.729 | 2.165 | 0.972 | 3.973 | 6.264 | 1.467 | 4.630 | 5.258 | 1.008 | 3.399 |
| | JS ↑ | $2.509 \times 10^{-2}$ | $1.365 \times 10^{-1}$ | $2.513 \times 10^{-3}$ | $1.354 \times 10^{-2}$ | $7.216 \times 10^{-2}$ | $1.386 \times 10^{-1}$ | $9.320 \times 10^{-3}$ | $9.080 \times 10^{-2}$ | $1.014 \times 10^{-1}$ | $3.977 \times 10^{-4}$ | **$6.130 \times 10^{-2}$** |

the extracted feature, the quantitative result is in Table 9 and the qualitative result is in Figure 4. We can find that in most cases there is a distinction between real and fake distribution. Furthermore, we show the detection performance of the classifier (following the setting as in Table 2) in the "TOFE with SD-V2.0" row of Table 7. We observe that the detection performance with Stable Diffusion v2.0 is comparable to that of Stable Diffusion v1.4, indicating that **the version of the pre-trained text-to-image model does not impact the results**.

Table 9: Quantitative result of features extracted by TOFE with Stable Diffusion v2.0.

| Metric | ADM | DALLE2 | DDPM | IDDPM | IF | LDM | PNDM | SD-V1 | SD-V2 | VQ-Diffusion | Average |
|---|---|---|---|---|---|---|---|---|---|---|---|
| MMD ↑ | 0.370 | 2.414 | 0.190 | 0.044 | 6.302 | 0.172 | 2.718 | 6.361 | 1.134 | 4.405 | 2.411 |
| JS ↑ | $3.841 \times 10^{-3}$ | $1.932 \times 10^{-2}$ | $6.392 \times 10^{-5}$ | $2.324 \times 10^{-5}$ | $1.353 \times 10^{-1}$ | $3.090 \times 10^{-4}$ | $3.819 \times 10^{-2}$ | $1.463 \times 10^{-1}$ | $3.663 \times 10^{-3}$ | $6.181 \times 10^{-2}$ | $4.089 \times 10^{-2}$ |

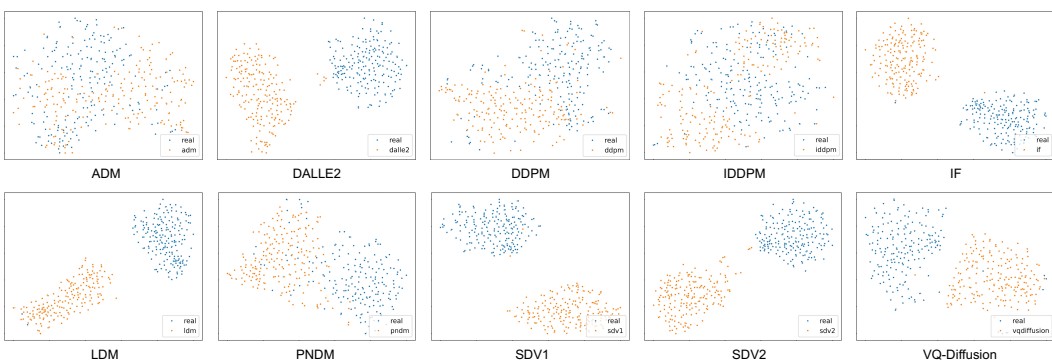

Figure 4: **Qualitative result of features extracted by TOFE with Stable Diffusion v2.0.** The visualization indicates that the TOFE method is unaffected by the version of the pre-trained guided Stable Diffusion model.

### A.4.3 ABLATION STUDY OF TOFE WITH $T = 50$

We demonstrate the representation achieved by TOFE when total timestep $T$=50. There are 50 refined representations and we take the representation at timestep $t$=1 for following qualitative and quantitative analysis. The quantitative result is in Table 10 and the qualitative result is in Figure 5. We can find that **in most cases there is a distinction between the real and fake distribution**.

Table 10: Quantitative result of features extracted by TOFE with $T = 50$.

| Metric | ADM | DALLE2 | DDPM | IDDPM | IF | LDM | PNDM | SD-V1 | SD-V2 | VQ-Diffusion | Average |
|---|---|---|---|---|---|---|---|---|---|---|---|
| MMD ↑ | 0.044 | 1.737 | 0.735 | 0.148 | 4.451 | 1.984 | 2.857 | 3.655 | 0.927 | 3.190 | 1.973 |
| JS ↑ | $3.675 \times 10^{-5}$ | $1.115 \times 10^{-2}$ | $8.000 \times 10^{-3}$ | $8.835 \times 10^{-4}$ | $7.252 \times 10^{-2}$ | $1.450 \times 10^{-2}$ | $3.278 \times 10^{-2}$ | $5.588 \times 10^{-2}$ | $5.368 \times 10^{-3}$ | $4.071 \times 10^{-2}$ | $2.418 \times 10^{-2}$ |

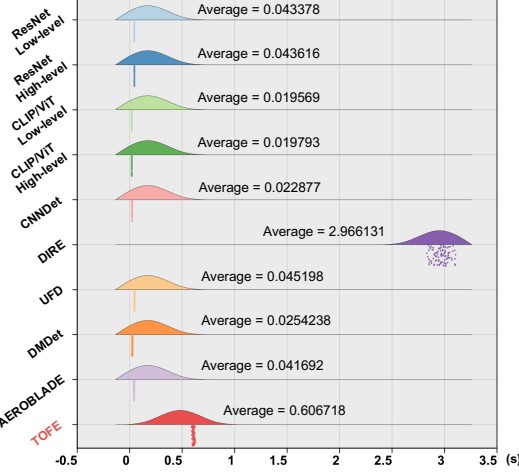

Figure 5: **Qualitative result of features extracted by TOFE with $T = 50$.** The visualization indicates that the TOFE method is unaffected by the number of timesteps used in feature extraction.

## A.5 TIME CONSUMPTION

Our TOFE method has a slight disadvantage in terms of time consumption. As shown in Figure 6, we evaluate the feature extraction procedure with ResNet50, CLIP-ViT-L-14, CNNDet, DIRE, UFD, DMDet, AEROBLADE and TOFE on 200 images, measuring the time taken per image. CLIP exhibited the fastest performance, with ResNet being slightly slower. Among the baselines and TOFE, DIRE was the slowest, taking nearly 3 seconds per image. CNNDet and DMDet require around 0.02 seconds, while UFD and AEROBLADE take about 0.4 seconds each. In comparison, the TOFE method is slightly slower, taking 0.6 seconds per image. However, as this is still under 1 second, the time consumption is acceptable, though we aim to reduce it in future work.

Figure 6: Time consumption of different feature extraction methods and baselines.

A.6 T-SNE Visualization of Various Feature Extraction Methods

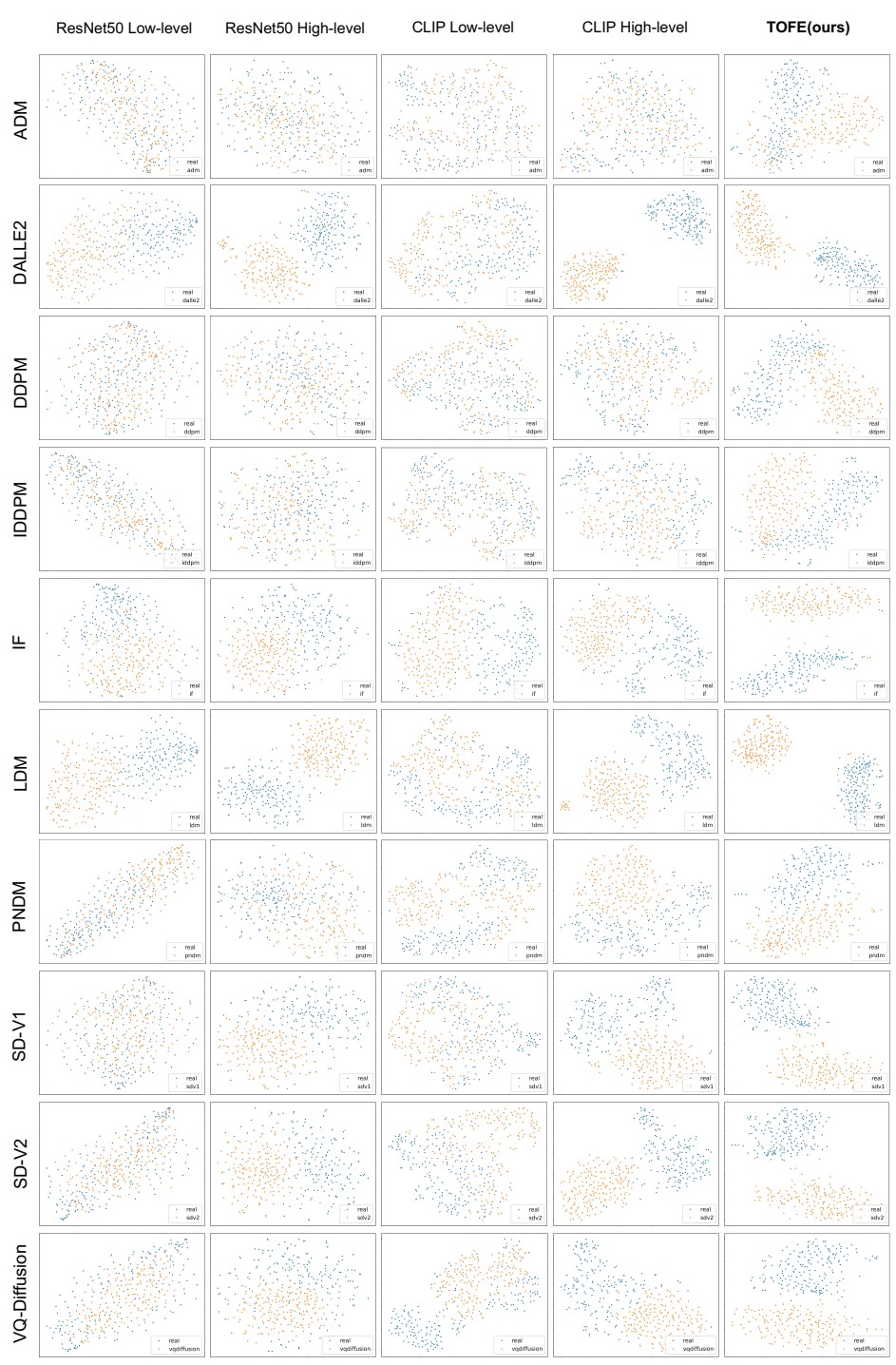

Figure 7: **T-SNE visualization of features extracted from ResNet, CLIP, and TOFE.** The visualization shows that real and fake images are distinctly separated based on TOFE features, highlighting the impressive performance of our method. Importantly, in types where ResNet or CLIP struggle to handle (*e.g.*, ADM, DDPM, IDDPM, PNDM).

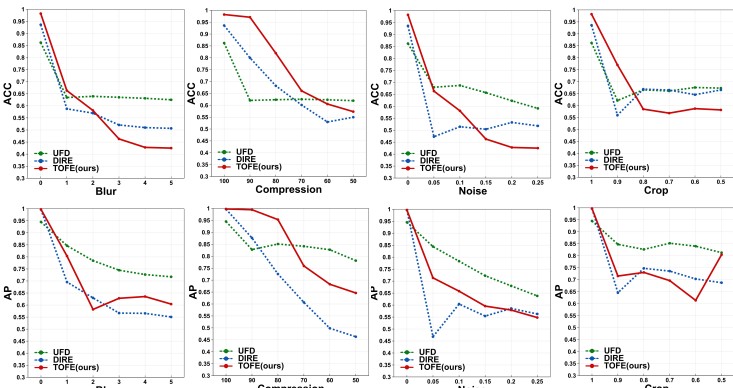

Figure 8: Robustness analysis under four corruptions.

## A.7 ROBUSTNESS ANALYSIS.

In real-world scenarios, images may face different common corruptions. Thus we evaluate how robust the TOFE is against common image corruptions. Referring to Frank et al. (2020); Ricker et al. (2024), we use Gaussian blur, JPEG compression, Gaussian noise, and Crop (with subsequent resizing to the original size). In Figure 8, we report the ACC and AP of TOFE for corruptions with different severities. For the four corruptions Gaussian blur / JPEG compression / Gaussian noise / Crop, following the setting of AEROBLADE Ricker et al. (2024), the severity standard deviation/compression quality/standard deviation/crop factor are 0,1,2,3,4,5 / 100,90,80,70,60,50 / 0,5,10,15,20,25 / 0.9,0.8,0.7,0.6,0.5. The result is averaged across ten diffusion types (types in Table 2). We can find that, in general, the ACC and AP of the classifier will reduce when corruption severity increases. The AP result under corruption of small severity is acceptable (higher than 70%) and values are still higher than 50% under corruption of high severity. For the comparison methods, we select a CNN-based detector, DIRE Wang et al. (2023), and a ViT-based detector, UFD Ojha et al. (2023). TOFE shows particular strength in mitigating corruption while a bit worse across other corruption types.

Here we give the visualization of corruption and their severity we used in robustness evaluation.

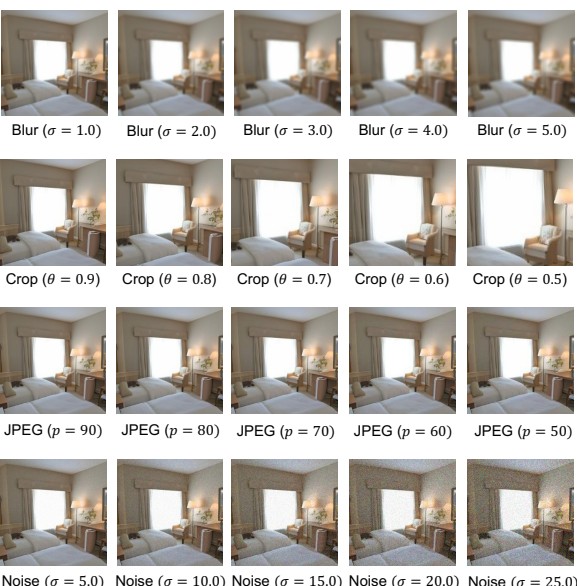

Figure 9: Visualization of different perturbations, including Gaussian noise, Gaussian blur, cropping, and JPEG compression.

## A.8 MULTIPLE METRICS EVALUATION

To comprehensively evaluate the proposed TOFE method for detecting DeepFake, we compared its performance with three other key metrics: F1 Score, Recall, and AUC.

**F1-Score** represents the harmonic mean of precision and recall, offering a balance between both metrics. As shown in Table 11, TOFE achieved an average F1 Score of 0.98, surpassing all baseline methods.

Table 11: Detection performance comparison with state-of-the-art detection methods (F1 Score).

| F1-Score | Diffusion Types | | | | | | | | | | Average |
|---|---|---|---|---|---|---|---|---|---|---|---|
| | ADM* | PNDM* | IDDPM* | DDPM | IF | LDM | DALLE2 | SD-V1 | SD-V2 | VQ-Diffusion | |
| CNNDet Wang et al. (2020) | 0.99 | 1.00 | 1.00 | 0.99 | 0.99 | 0.99 | 0.77 | 0.95 | 0.72 | 0.98 | 0.94 |
| UFD Ojha et al. (2023) | 0.90 | 0.95 | 0.99 | 0.98 | 0.21 | 0.51 | 0.98 | 0.80 | 0.76 | 0.98 | 0.81 |
| DIRE Wang et al. (2023) | 0.86 | 0.94 | 0.89 | 0.98 | 0.94 | 0.86 | 0.96 | 0.99 | 0.91 | 0.92 | 0.93 |
| DMDet Corvi et al. (2023) | 0.99 | 1.00 | 1.00 | 1.00 | 0.51 | 0.94 | 0.14 | 0.51 | 0.13 | 0.98 | 0.72 |
| TOFE (ours) | 0.97 | 0.99 | 0.99 | 0.98 | 0.99 | 0.98 | 0.97 | 0.98 | 0.97 | 0.99 | **0.98** |

**Recall** measures the ability to identify all positive instances correctly. As shown in Table 12, TOFE demonstrated excellent detection capability, with an average Recall of 0.98. It achieved near-perfect results (Recall = 1.00) on datasets like PNDM and VQ-Diffusion, showing minimal false negatives and high reliability.

Table 12: Detection performance comparison with state-of-the-art detection methods (Recall).

| Recall | Diffusion Types | | | | | | | | | | Average |
|---|---|---|---|---|---|---|---|---|---|---|---|
| | ADM* | PNDM* | IDDPM* | DDPM | IF | LDM | DALLE2 | SD-V1 | SD-V2 | VQ-Diffusion | |
| CNNDet Wang et al. (2020) | 0.99 | 1.00 | 1.00 | 1.00 | 0.99 | 1.00 | 0.63 | 0.91 | 0.57 | 0.97 | 0.91 |
| UFD Ojha et al. (2023) | 0.84 | 0.94 | 1.00 | 0.99 | 0.12 | 0.35 | 1.00 | 0.69 | 0.63 | 1.00 | 0.75 |
| DIRE Wang et al. (2023) | 0.76 | 0.90 | 0.81 | 0.97 | 0.90 | 0.76 | 0.94 | 0.98 | 0.84 | 0.85 | 0.87 |
| DMDet Corvi et al. (2023) | 0.99 | 1.00 | 1.00 | 1.00 | 0.35 | 0.88 | 0.07 | 0.34 | 0.07 | 0.98 | 0.67 |
| TOFE (ours) | 0.96 | 1.00 | 0.99 | 0.98 | 0.98 | 0.98 | 0.97 | 0.97 | 0.95 | 1.00 | **0.98** |

**AUC** reflects the overall ability of the model to distinguish between classes across all thresholds. As shown in Table 13, TOFE exhibits exceptional performance in terms of AUC-ROC, achieving an average score of 0.998, which is on par with DIRE and slightly exceeds DMDet in certain cases.

Table 13: Detection performance comparison with state-of-the-art detection methods (AUC).

| AUC | Diffusion Types | | | | | | | | | | Average |
|---|---|---|---|---|---|---|---|---|---|---|---|
| | ADM* | PNDM* | IDDPM* | DDPM | IF | LDM | DALLE2 | SD-V1 | SD-V2 | VQ-Diffusion | |
| CNNDet Wang et al. (2020) | 1.000 | 1.000 | 1.000 | 1.000 | 1.000 | 1.000 | 0.976 | 0.996 | 0.960 | 0.999 | 0.993 |
| UFD Ojha et al. (2023) | 0.978 | 0.989 | 0.999 | 0.999 | 0.800 | 0.857 | 0.999 | 0.961 | 0.937 | 0.999 | 0.952 |
| DIRE Wang et al. (2023) | 0.996 | 0.999 | 0.997 | 1.000 | 0.998 | 0.995 | 0.999 | 1.000 | 0.998 | 0.998 | **0.998** |
| DMDet Corvi et al. (2023) | 1.000 | 1.000 | 1.000 | 1.000 | 0.981 | 0.999 | 0.991 | 0.995 | 0.995 | 1.000 | 0.996 |
| TOFE (ours) | 0.997 | 1.000 | 0.999 | 0.998 | 0.999 | 0.998 | 0.995 | 0.998 | 0.995 | 1.000 | **0.998** |

Overall, the experimental results demonstrate that TOFE achieves superior performance compared to existing methods across all three metrics highlighting its effectiveness in detecting diffusion-based DeepFakes. Importantly, **TOFE exhibits remarkable out-of-domain generalization capability across diverse diffusion types**, including LDM, DDPM, and SD-V1/V2, among others. This suggests that the TOFE model effectively captures intrinsic differences between real and synthesized images, irrespective of the underlying generation mechanism. By focusing on essential image features, TOFE extracts robust representations that align with the fundamental characteristics of genuine and manipulated content, ensuring its consistent and reliable performance across varied DeepFake datasets. These results underline TOFE's potential as a generalized and versatile solution for addressing the challenges of DeepFake detection.

