# OpenReview forum: "Text Modality Oriented Image Feature Extraction for Detecting Diffusion-based DeepFake"
_ICLR.cc/2026/Conference — Submitted to ICLR 2026_

### Official Review · Reviewer_brHr · 2025-10-29

**Soundness:** 3
**Presentation:** 3
**Contribution:** 3
**Rating:** 4
**Confidence:** 4

**Summary:**

This paper introduces a novel cross-modal approach to DeepFake detection. Specically, this paper proposes TOFE, a method that transforms image information into the text embedding space. The authors argue that the the optimized text embedding contain both high-level and low-level features of the given images. Such features can help the detectors better capture the information of fake and real images. The authors have conducted abundent experiments to demonstrate that the proposed methods can work well on different settings compared to other state-of-the-art methods.

**Strengths:**

- Novel methods for important research questions
- Well-written paper
- Sufficient experimental evidence

**Weaknesses:**

- Lack of theoretical support
- Lack of more practical settings

**Questions:**

This paper proposed the novel fake image detection methods by combining both high-level features and low-level features into the optimized text embeddings which can help the detectors better distinguish fake images apart from real ones. Experimental results demonstrate that the proposed methods can have great performance on the DIRE datasets than other methods. The paper is well-written and the experimental results are convincing. However, I do have some comments regarding the theoretical support and the practical settings in this paper.

- The most important questions is why the optimized text can help boost the detection performance. The authors are supposed to give more explanations on why TOFE works.

- The authors trained and tested the proposed methods on the DIRE datasets, which is constructed in 2023. In recent years, there have been many developments in the field of diffusion generation. The authors are supposed to test their methods on the state-of-the-art generation models like Stable Diffusion 3.5, Qwen-Image-Edit, and the latest version of Dalle. The current results(99 aacuracy) cannot reflect the challenge of identifying fake images in real-world scenarios.

- The authors are also supposed to argue the performance of the proposed methods on the edited images as well as the adversary examples that are designed to confuse the detectors. Moreover, the proposed methods may be too time-consuming and computationally expensive for practical deployment, especially when scaling to large datasets or real-time detection scenarios. The authors should therefore provide an analysis or discussion of the efficiency trade-offs and potential optimizations to make their approach more feasible in real-world applications.

---

> ### Author Response · Authors · 2025-11-25
> **(1/3) Rebuttal from Authors of Submission12427 to Reviewer brHr**
>
> We are sincerely grateful for your valuable feedback and the careful attention you devoted to reviewing our work.  Our point-by-point response to your concerns is provided below:
>
> > Question 1: "The most important questions is why the optimized text can help boost the detection performance. The authors are supposed to give more explanations on why TOFE works."
> ***
>
> **Response:**
> We answer this question from three perspectives: the precise meaning of the “text” in TOFE, the information it represents, and how this information enhances detection performance.
>
> ### 1. What is the text embedding?
>
> In our method, we directly **optimize** the **text embedding**, not natural-language text. The **optimized continuous text embedding** ($\hat{\mathcal{C}}$) is a high-dimensional vector in the text-embedding space of a pre-trained text-to-image model. This embedding is optimized to precisely encode the visual information of a target image, enabling the T2I model to reconstruct the image’s latent representation accurately. After this optimization, $\hat{\mathcal{C}}$ serves as the feature representation used for DeepFake detection.
>
> ### 2. What does the text embedding represent?
>
> The optimized embedding $\hat{\mathcal{C}}$ simultaneously captures **high-level semantic information** and **low-level texture details**. Through the reconstruction-driven optimization process, the embedding is updated to minimize latent-space reconstruction error. This forces $\hat{\mathcal{C}}$ to encode semantic information, fine textures, and subtle generative artifacts. Consequently, the embedding becomes a compact and information-rich representation that contains both **high-level** and **low-level features**.
>
>
> ### 3. Why is it effective for deepfake detection?
>
> The optimized embedding enhances detection performance because it constructs a **highly discriminative, functionally fused** feature space. The reconstruction objective inherently integrates high-level semantics with low-level fidelity: structural coherence must be preserved, and fine-grained details must be matched at pixel and latent levels. This joint encoding sharply increases the separability between real and fake image distributions, enabling TOFE to achieve strong detection performance across diverse diffusion models, even with a simple classifier.

---

> ### Author Response · Authors · 2025-11-25
> **(2/3) Rebuttal from Authors of Submission12427 to Reviewer brHr**
>
> > Question 2: "The authors trained and tested the proposed methods on the DIRE datasets, which is constructed in 2023. In recent years, there have been many developments in the field of diffusion generation. The authors are supposed to test their methods on the state-of-the-art generation models like Stable Diffusion 3.5, Qwen-Image-Edit, and the latest version of Dalle. The current results(99 aacuracy) cannot reflect the challenge of identifying fake images in real-world scenarios."
> ***
>
> **Response:**
>
> We appreciate the reviewer's insightful comment on the importance of evaluating DeepFake detection against emerging models. The generalizability of detection methods is indeed a crucial metric given the rapid evolution of generative architectures.
>
> We conducted a dedicated **transferability experiment** to evaluate TOFE's generalization capability under a more **realistic and challenging scenario**. The model was trained exclusively on data from three diffusion models (ADM, PNDM, IDDPM) and then tested on five unseen, state-of-the-art models, simulating a true cross-model transfer setting.
>
> The transferability results show a clear divergence:
>
> * **DiT Architectures:** TOFE achieves high detection accuracy on **Dalle3** (ACC ≈ 0.96%), demonstrating robust generalization within the Diffusion Transformer family.
> * **Flow Matching Models:** Performance drops sharply on SD3.5 and FLUX.1-dev. Overall ACC falls to ≈ 0.50%, with Fake Image Accuracy (F_ACC) between 1.92% and 5.42%.
>
> **Table 1: TOFE Feature Detection Performance Summary Across Generative Models (Cross-Model Transfer)**
> | Model Name       | Generation Method | ACC (%) | AP (%) | R_ACC (%) | F_ACC (%) | F1 (%) | ROC (%) |
> | :--------------- | :---------------- | ------: | -----: | --------: | --------: | -----: | ------: |
> | Dalle3           | Closed-Source     | 96.36   | 93.10  | 98.60     | 81.13     | 84.94  | 98.80  |
> | Hunyuan-DiT          | Diffusion         | 85.67   | 98.02  | 98.60     | 72.65     | 83.51  | 98.20  |
> | SD3.5            | Flow Matching     | 51.99   | 68.00  | 98.61     | 5.42      | 10.10  | 68.56  |
> | FLUX.1-dev       | Flow Matching     | 50.23   | 59.84  | 98.60     | 1.92      | 3.72   | 64.72  |
> | Qwen-Image-Edit  | Diffusion         | 78.79   | 91.78  | 98.57     | 52.94     | 68.37  | 91.28  |
>
> This decline highlights a challenge in feature compatibility across generative paradigms. TOFE relies on the **LDM manifold** to capture low-level artifacts from DDPM/DDIM processes. Flow Matching models, however, generate continuous probability flows that produce different low-level distributions, to which the LDM-based encoder is insensitive. These findings indicate that while TOFE generalizes well within certain architectures, adapting to Flow Matching models requires further research on cross-paradigm feature representations.

---

> ### Author Response · Authors · 2025-11-25
> **(3/3) Rebuttal from Authors of Submission12427 to Reviewer brHr**
>
> > Question 3: "The authors are also supposed to argue the performance of the proposed methods on the edited images as well as the adversary examples that are designed to confuse the detectors. Moreover, the proposed methods may be too time-consuming and computationally expensive for practical deployment, especially when scaling to large datasets or real-time detection scenarios. The authors should therefore provide an analysis or discussion of the efficiency trade-offs and potential optimizations to make their approach more feasible in real-world applications."
> ***
>
> **Response:**
>
> ### 1. Robustness to Edited Images
> We tested TOFE's performance using **Qwen-Image-Edit** to modify real images. The results showed that TOFE achieved an overall accuracy (ACC) of **78.79%** and an average precision (AP) of **91.78%**, indicating a mid-to-high level of capability in identifying this type of edited image.
>
> **Table 2: Detection Performance on Edited Images**
>
> | Model Name       | Method      | ACC (%) | AP (%) | R_ACC (%) | F_ACC (%) | F1 (%) | ROC (%) |
> | :--------------- | :---------- | ------: | -----: | --------: | --------: | -----: | ------: |
> | Qwen-Image-Edit  | Diffusion-based | 78.79   | 91.78  | 98.57     | 52.94     | 68.37  | 91.28  |
>
> ### 2. Performance on Adversarial Examples Aimed at Fooling Detectors
>
> We evaluated TOFE under a practical adversarial setting, where adversarial examples were generated using PGD attacks with varying perturbation strengths (ε). The attacks were applied to the UFD detector proposed by Ojha et al. (2023) [1], which served as the target model during optimization. Across all ε values, the PGD attacks completely collapsed the UFD detector, reducing both its accuracy (ACC) and average precision (AP) to **0%**. TOFE, using only a simple classifier (i.e., MLP), **maintained acceptable detection performance** on these adversarial examples, demonstrating effective resistance to transfer-based white-box attacks.
>
> Reference:
>
> [1] Utkarsh Ojha, Yuheng Li, and Yong Jae Lee. *Towards universal fake image detectors that generalize across generative models.*  CVPR 2023.
>
> **Table 3: Performance of Detector under $\text{L}_{\infty}$ PGD Attack**
>
> | Epsilon ($\epsilon$) | ACC (%) | AP (%) | F_ACC (%) | R_ACC (%) | ROC (%) |
> | :---: | :---: | :---: | :---: | :---: | :---: |
> | $1/255$ | 61.94 | 65.33 | 43.94 | 79.98 | 0.6716 |
> | $2/255$ | 58.76 | 62.20 | 37.34 | 80.20 | 0.6346 |
> | $4/255$ | 56.93 | 61.15 | 33.49 | 80.36 | 0.6257 |
>
>
> ### 3.Efficiency Trade-offs and Optimization
>
> We appreciate the reviewer's concern regarding the computational cost of TOFE for real-world deployment. TOFE's current feature extraction time is **0.607 seconds per image**. We have already achieved an acceleration of approximately **4.89 times** compared to the DIRE [1] detection method (which takes 2.966 seconds per image), referencing Appendix A.5, Figure 6.
>
> We have explored efficiency optimization methods, specifically reducing the sampling steps from the theoretical **T=50** to **T=1**. The T=1 setting not only provided faster speed but also yielded superior detection performance, achieving an average ACC of 98.21% and AP of 99.76%. We believe this speed of **approximately 0.6 seconds per image is acceptable** and has the potential to scale to larger datasets.
>
> Looking ahead, we will explore more efficient inversion schemes, investigating faster and more efficient latent space inversion methods than DDIM to meet the needs of real-time and large-scale detection scenarios.
>
> References:
>
> [1] Zhendong Wang, Jianmin Bao, Wengang Zhou, Weilun Wang, Hezhen Hu, Hong Chen, and Houqiang Li. *DIRE for diffusion-generated image detection.* CVPR 2023.

---

### Official Review · Reviewer_c3dr · 2025-10-30

**Soundness:** 2
**Presentation:** 3
**Contribution:** 2
**Rating:** 6
**Confidence:** 3

**Summary:**

The paper proposes TOFE (Text‑modality Oriented Feature Extraction) for detecting diffusion‑generated DeepFakes. Instead of extracting features purely in the image domain, TOFE refines continuous text embeddings so that, under DDIM inversion and classifier‑free guidance, they reproduce the target image’s reverse trajectory; the refined embeddings are then fed to a small MLP classifier.

**Strengths:**

1. The paper moves feature extraction to the text modality: starting from a text embedding C, it optimizes C so that conditional denoising follows the inverted latent trajectory of the target image. Figures 2–3 illustrate the setup and what the optimization is correcting.

2. On DIRE (10 diffusion generators), TOFE attains the best average ACC/AP (98.21% / 99.76%) against other methods.

**Weaknesses:**

1. Quantifying separability on 2D t‑SNE embeddings risks unreliable distance comparisons. T‑SNE‑then‑MMD/JS analysis may distort distances.

2. Figure 2 uses BLIP semantic initialization as an example, but author implements the default empty prompt embedding as the starting point, and the impact of the difference between the two on the final feature/detection effect is not systematically presented.

**Questions:**

1. What is the impact of null vs. caption‑initialized embeddings (e.g., BLIP) on both reconstruction quality ?

---

> ### Author Response · Authors · 2025-11-25
> **(1/2) Rebuttal from Authors of Submission12427 to Reviewer c3dr**
>
> We are sincerely grateful for your valuable feedback and the careful attention you devoted to reviewing our work.  Our point-by-point response to your concerns is provided below:
>
> > Weakness 1: "Quantifying separability on 2D t‑SNE embeddings risks unreliable distance comparisons. T‑SNE‑then‑MMD/JS analysis may distort distances."
> ***
>
> **Response:**
> We appreciate you raising this valid critique regarding the potential for t-SNE embeddings to distort distances and affect separability quantification.
>
> To address this, we recomputed the Maximum Mean Discrepancy (MMD) directly on the high-dimensional TOFE features before applying any dimensionality reduction. These high-dimensional results confirm the separability advantage holds: the average **MMD value for TOFE features (0.7883)** is significantly superior to the average MMD values achieved by ResNet and CLIP across their low-level, high-level, and concatenated feature combinations, thus validating the intrinsic separability shown in our original visualizations.
>
> Maximum Mean Discrepancy ($\text{MMD} \uparrow$) values for feature spaces extracted by ResNet-50, CLIP ViT-L/14, and TOFE. Higher $\text{MMD}$ indicates a greater feature separability.
>
> **Table 1: MMD Distance Comparison of Feature Extraction Methods Across Diffusion Models**
>
> |  | Method | ADM | DALLE2 | DDPM | IDDPM | IF | LDM | PNDM | SD-V1 | SD-V2 | VQ-Diffusion | Average |
> | :---: | :---: | :---: | :---: | :---: | :---: | :---: | :---: | :---: | :---: | :---: | :---: | :---: |
> | **ResNet-50** | Low-level | 0.0556 | 0.1802 | 0.0837 | 0.0696 | 0.2760 | 0.4209 | 0.2779 | 0.1073 | 0.1334 | 0.1520 | 0.1757 |
> | | High-level | 0.0100 | 0.1057 | 0.0204 | 0.0156 | 0.0977 | 0.1472 | 0.0491 | 0.0806 | 0.1010 | 0.0834 | 0.0711 |
> | | Low&High | 0.0163 | 0.1166 | 0.0299 | 0.0234 | 0.1225 | 0.1883 | 0.0862 | 0.0831 | 0.1045 | 0.0919 | 0.0863 |
> | **CLIP ViT-L/14** | Low-level | 0.1458 | 0.2672 | 0.2073 | 0.1582 | **1.4098** | 0.3191 | 0.7922 | 0.4779 | **0.9004** | 0.8450 | 0.5523 |
> | | High-level | **0.2171** | 0.5101 | 0.2369 | **0.2615** | 0.8558 | 0.9466 | 0.5736 | 0.5010 | 0.8133 | 0.7812 | 0.5697 |
> | | Low&High | 0.1507 | 0.3023 | 0.2064 | 0.1644 | 1.3483 | 0.4118 | 0.7684 | 0.4823 | 0.8908 | 0.8450 | 0.5570 |
> | **TOFE (Ours)** | **Combined** | 0.1594 | **0.9852** | **0.2446** | 0.2549 | 0.8757 | **1.0880** | **1.5355** | **0.9678** | 0.8914 | **0.8809** | **0.7883** |
>
> We focused exclusively on MMD because estimating probability density functions (PDFs) for computing Jensen–Shannon (JS) divergence becomes highly unstable in extremely high-dimensional feature spaces  [1], making JS computation impractical.
>
> References:
>
> [1] Litvinenko, A., Khoromskij, B., & Matthies, H. G., "Computing f‑Divergences and Distances of High‑Dimensional Probability Density Functions — Low‑Rank Tensor Approximations," arXiv 2021.

---

> ### Author Response · Authors · 2025-11-25
> **(2/2) Rebuttal from Authors of Submission12427 to Reviewer c3dr**
>
> > Weakness 2: "Figure 2 uses BLIP semantic initialization as an example, but author implements the default empty prompt embedding as the starting point, and the impact of the difference between the two on the final feature/detection effect is not systematically presented."
> ***
>
> **Response:**
> We thank the reviewer for the comment. We compared **BLIP semantic initialization** with our default **empty prompt (null-text) initialization**. The results show that features extracted using null-text consistently achieve higher Maximum Mean Discrepancy (MMD) values and lead to better deepfake detection performance, whereas BLIP-initialized features exhibit lower MMD and reduced detection accuracy.
>
> The lower discriminability of BLIP-initialized features can be attributed to the semantic biases encoded in the BLIP embeddings, which may partially misalign with the intrinsic latent structure of the images. In contrast, null-text initialization provides a clean latent, yielding features that **better capture the image characteristics** and are more effective for deepfake detection.
>
> **Table 2: MMD Comparison of Feature Spaces Extracted Using BLIP Initialization and TOFE**
>
> | Feature Space | Metric | ADM | DALLE2 | DDPM | IDDPM | IF | LDM | PNDM | SD-V1 | SD-V2 | VQ-Diffusion | Average |
> | :---: | :---: | :---: | :---: | :---: | :---: | :---: | :---: | :---: | :---: | :---: | :---: | :---: |
> | **BLIP Init** | MMD | 0.0772 | 0.3882 | 0.1160 | 0.1042 | 0.1601 | 0.1990 | 0.1059 | 0.2102 | 0.2568 | 0.3062 | 0.1924 |
> | **TOFE: Null Text** | MMD | **0.1594** | **0.9852** | **0.2446** | **0.2549** | **0.8757** | **1.0880** | **1.5355** | **0.9678** | **0.8914** | **0.8809** | **0.7983** |
>
>
> Table 3 summarizes the detection performance of a BLIP-initialized feature extraction across ten generative models, achieving an overall average **accuracy of 67.47%** and **average precision of 69.09%**.
>
> **Table 3: Detection Performance of BLIP-Initialized Feature Extraction Across Generative Models**
>
> | **Models** | **Accuracy (%)** | **Average Precision (%)** | **F1 Score (%)** | **ROC AUC (%)** |
> | :--- | :--- | :--- | :--- | :--- |
> | **ADM** | 65.21 | 69.06 | 67.55 | 71.39 |
> | **DALLE2** | 66.15 | 61.38 | 61.83 | 77.79 |
> | **DDPM** | 66.97 | 68.85 | 67.42 | 76.69 |
> | **IDDPM** | 69.74 | 73.74 | 72.89 | 76.79 |
> | **IF** | 54.25 | 58.95 | 52.49 | 57.24 |
> | **LDM** | 69.24 | 73.71 | 72.35 | 76.26 |
> | **PNDM** | 66.01 | 69.61 | 68.47 | 72.17 |
> | **SD-V1** | 65.00 | 70.08 | 67.17 | 72.09 |
> | **SD-V2** | 68.60 | 70.53 | 71.63 | 75.27 |
> | **VQ-Diffusion** | 70.58 | 75.05 | 73.90 | 79.72 |
> | **Average** | **67.47** | **69.09** | **67.77** | **73.54** |
>
>
> > Question 1: "What is the impact of null vs. caption‑initialized embeddings (e.g., BLIP) on both reconstruction quality ?"
> ***
>
> **Response:**
> We appreciate the inquiry regarding the influence of initialization methods on reconstruction quality. We compared BLIP semantic initialization with our null-text (TOFE) approach, employing $\text{PSNR}$ and $\text{SSIM}$ as quantitative metrics.
>
> Our findings reveal that features derived from null-text initialization consistently yield **superior reconstruction fidelity** across diverse generative models, evidenced by generally higher $\text{PSNR}$ and $\text{SSIM}$ values compared to BLIP initialization.
>
> This improved performance is predicated on the **null-text strategy providing an unconditioned latent reference**. By avoiding the semantic bias introduced by caption-initialized embeddings, the extracted features better capture the **intrinsic structural and textural details** of the image. This comprehensive representation is essential for minimizing information loss and achieving **high-fidelity reconstruction**.
>
> **Table 4: PSNR ($\uparrow$) and SSIM ($\uparrow$) values for reconstructed images using BLIP initialization and TOFE (Null Text)**
>
> | **Metric**                | **LDM**  | **IF**   | **VQ-Diff** | **DDPM**  | **ADM**   | **IDDPM** | **Dalle2** | **PNDM**  | **SD-V1** | **SD-V2** | **Real**  |
> | :-------------------------------- | -------: | -------: | ---------------: | --------: | --------: | --------: | ----------: | --------: | --------: | --------: | --------: |
> | **PSNR ↑ (Blip Init)**            | 37.4193  | 35.8437  | 34.3039          | 33.9041   | 34.0212   | 33.3816   | 32.6942     | 30.6984   | 31.7374   | 31.0123   | 31.8676   |
> | **SSIM ↑ (Blip Init)**            | 0.9703   | 0.9643   | 0.9376           | 0.9329    | 0.9315    | 0.9204    | 0.9058      | 0.8583    | 0.9309    | 0.8562    | 0.9028    |
> | **PSNR ↑ (TOFE: Null Text)**      | **41.485** | **38.754** | **36.190**     | **35.618** | **35.391** | **35.001** | **33.808** | **31.678** | 30.025    | **32.203** | 31.486    |
> | **SSIM ↑ (TOFE: Null Text)**      | **0.977**  | **0.970**  | **0.943**      | **0.942**  | **0.935**  | **0.927**  | **0.911**  | **0.871**  | 0.874     | **0.873**  | 0.891     |

---

### Official Review · Reviewer_S9Kt · 2025-10-31

**Soundness:** 1
**Presentation:** 2
**Contribution:** 1
**Rating:** 2
**Confidence:** 5

**Summary:**

This paper proposes a method for detecting fake images. The central claim is that both low-level and high-level features are crucial for effective fake image detection. The authors further hypothesize that as the quality of generated images improves, it becomes increasingly difficult to detect them using only domain specific visual features. Their key insight is that the text embeddings used to guide image generation, containing both low and high-level semantic information exhibit discriminative properties well-suited for fake image detection. Experimental results demonstrate consistent improvements over several baselines on images generated by both diffusion models and GANs.

**Strengths:**

1. The TOFE method presented by the paper outperforms some popular baselines.

**Weaknesses:**

Major Weaknesses
1. Line 43: The paper discusses moving past the "traditional binary classification" mindset. However, binary classification is simply the definition of the problem, and TOFE also does the same using different features, therefore the motivation seems unclear to me. The paper would greatly benefit from greater clarity here.
2. The main argument regarding the necessity of both high-level and low-level features needs more evidence. For instance, are the authors able to identify specific low/high-level features. The current experiments (such as Section 3), are done on ResNet and CLIP backbones. While the effectiveness of CLIP for fake image detection has been shown in the past, there are several other techniques (training the network for fake image detection, reconstruction based detection), and it is not clear to me that these methods suffer from the same drawbacks.
3. Line 62-63: distinguishing between real and generated images within the image domain is becoming increasingly challenging. This claim is not entirely true. While generalizing to other generators/unseen post-processing operations remain challenging, usually trained classifiers are able to discriminate between real and fake distributions successfully. I believe this statement needs to be worded with further clarity, since it is critical to the design choice.
4. line 79-80: This is also not entirely true. There are other factors of variation in a diffusion model such as starting noise and added randomness (DDPM style). Therefore, it is difficult to attribute a single representation for a given image. The representation might incorporate important features (as shown by [1]), but it is not clear to me if the text will steer the generation towards a "single target image".
5. line 267-269: This does not explain why only low level features are not the right way to go. Additionally, in fact [2] show that if post-processing is applied during training (getting rid of some low-level features), the model generalizes better, which seems contrary to the hypothesis being presented here. I believe the paper would benefit by being more clear here.
6. Line 305-306 talks about how any representation with high-level and low-level features is not necessarily helpful. This raises doubts on what is the thing that is special about the representations obtained by TOFE. It would be helpful if the authors can conduct experiments to understand this in a deeper manner.
7. The paper currently lacks experiments studying the sensitivity to post-processing. Conducting a sweep over JPEG/Resizing (both upsampling and downsampling), WEBP and blur would make the work more suited for practical applications.

Minor Weaknesses/Suggestions,
1. Line 41: If the main task is fake image detection, what is the downstream task.
2. Line 42: Ojha et al is designed for universal fake image detection (they test on GANs as well, not only diffusion)
3. Line 224-225: DDIM is not the only way to perform denoising (DDPM + other solvers such as PNDM would also work).
4. Line 42: DIRE is cited as a feature extraction method, but it is reconstruction-based and it would be better if it was distinguished from other feature extraction methods).
5. Sec 4.2.2 is titled Problem Definition, but it talks about the particular algorithm developed by the authors. Problem Definition should talk about the problem which the authors are trying to solve.

References
1. Gal, R., Alaluf, Y., Atzmon, Y., Patashnik, O., Bermano, A. H., Chechik, G., & Cohen-Or, D. (2022). An image is worth one word: Personalizing text-to-image generation using textual inversion. arXiv preprint arXiv:2208.01618.
2. Wang, S. Y., Wang, O., Zhang, R., Owens, A., & Efros, A. A. (2020). CNN-generated images are surprisingly easy to spot... for now. In Proceedings of the IEEE/CVF conference on computer vision and pattern recognition (pp. 8695-8704).

**Questions:**

1. Is the Image being DDIM inverted using the unconditional diffusion model? If so why?
2. Related to above, it seems to me like the DDIM sampling is done from the unconditionally inverted latent. Why is the embedding being optimized to align with the unconditional trajectory. One possible solution to the same could be the null-text embedding (which would mimic the original trajectory). Please clarify the motivation behind this method.
3. Algorithm 1: It starts by saying z_0 is obtained from the image (seems like it comes from the latent encoder). Then it proceeds to say that z_0 is computed from DDIM inversion which is confusing to me.
4. Line 299-302 talks about extracting features at every time step. However, only T=1 is being used in line 350-351. This design choice would benefit from stronger motivation.

---

> ### Author Response · Authors · 2025-11-25
> **(1/4) Rebuttal from Authors of Submission12427 to Reviewer S9Kt**
>
> We are sincerely grateful for your valuable feedback and the careful attention you devoted to reviewing our work.  Our point-by-point response to your concerns is provided below:
>
> >  Major Weakness 1: "Line 43: The paper discusses moving past the "traditional binary classification" mindset. However, binary classification is simply the definition of the problem, and TOFE also does the same using different features, therefore the motivation seems unclear to me. The paper would greatly benefit from greater clarity here."
> ***
>
> **Response:**
> We apologize for the ambiguity in the statement "treating the detection task as a traditional binary classification problem". We fully acknowledge that DeepFake detection is fundamentally a binary classification task.
>
> Our intended point was to highlight a limitation in the feature extraction focus of existing methods. As discussed in our introduction, prior approaches often perform feature extraction similarly to generic image classification tasks. This conventional mindset primarily prioritizes high-level semantic features while often overlooking low-level features. However, these low-level details are critical for identifying the subtle artifacts of diffusion models.
>
> Therefore, our motivation is not to redefine the binary classification problem, but to introduce a feature extraction paradigm (TOFE) that explicitly captures both high-level semantics and low-level details. We will revise the manuscript to clarify that our contribution lies in ensuring the feature representation is comprehensive, preventing any misunderstanding regarding the task definition.
>
> > Major Weakness 2: "The main argument regarding the necessity of both high-level and low-level features needs more evidence. For instance, are the authors able to identify specific low/high-level features. The current experiments (such as Section 3), are done on ResNet and CLIP backbones. While the effectiveness of CLIP for fake image detection has been shown in the past, there are several other techniques (training the network for fake image detection, reconstruction based detection), and it is not clear to me that these methods suffer from the same drawbacks."
> ***
>
> **Response:**
> We thank the reviewer for raising this important point. Below, we explicitly identify the specific low/high-level features used in our current experiments and provide more evidence regarding the necessity of both feature levels.
>
> ### 1. Identification of Specific Low/High-level Features
> As detailed in **Appendix A.3**, we identify the specific features based on the ResNet and CLIP backbones used in our experiments:
>
> Low-level Features: We identify these as the output from the shallow layers (the output of the first block), which capture texture and artifacts.
> * For ResNet: We use the output of `layer1` (dimension: $1\times256\times56\times56$).
> * For CLIP: We use the output of `resblocks0` (dimension: $257\times1\times1024$).
>
> High-level Features: We identify these as the output from the deep layers (the output of the last layer), which represent semantics.
> * For ResNet: We use the output of the `average pooling` layer (dimension: $1\times2048\times1\times1$).
> * For CLIP: We use the output of the `layernorm` layer after `resblocks11` (dimension: $1\times768$).
>
> ### 2. Evidence for the Necessity of Both High-level and Low-level Features
> To support the main argument regarding the necessity of both high-level and low-level features, we cite related research showing that single-level features are insufficient:
>
> * Cheng et al. (CVPR 2025) [1] explicitly categorize current detection paradigms into "Semantic-based" (High-level) and "Texture/Pixel-based" (Low-level). They demonstrate that purely semantic methods miss pixel-level forgery traces, while purely texture-based methods fail on semantically inconsistent but texturally perfect generations. They prove that combining semantic and pixel features is essential to detect the latest AI-generated images.
>
> * Wang et al. (arXiv 2025) [2] highlight that simple combinations can lead to noise amplification. They argue for a synergy between "Local Artifacts" (Low-level) and "Mesoscopic Semantics" (High-level) to achieve robust detection. This aligns with our finding in Table 3 that simple concatenation ("Low & High") is unstable, whereas our TOFE method—which implicitly fuses these features via text embedding optimization—achieves superior performance.
>
> References:
>
> [1] Cheng et al., "CO-SPY: Combining Semantic and Pixel Features to Detect Synthetic Images by AI", CVPR 2025.
>
> [2] Wang et al., "Morphology-optimized Multi-Scale Fusion: Combining Local Artifacts and Mesoscopic Semantics", arXiv 2025.

---

> ### Author Response · Authors · 2025-11-25
> **(2/4) Rebuttal from Authors of Submission12427 to Reviewer S9Kt**
>
> >  Major Weakness 3: "line 79-80: This is also not entirely true. There are other factors of variation in a diffusion model such as starting noise and added randomness (DDPM style). Therefore, it is difficult to attribute a single representation for a given image. The representation might incorporate important features (as shown by [1]), but it is not clear to me if the text will steer the generation towards a "single target image"."
> ***
>
> **Response:**
> We agree with the reviewer that diffusion models are inherently stochastic, and we acknowledge that our phrasing "single representation" was imprecise.
>
> We would like to clarify that we used this term to describe the **validity of the reconstruction**. Our intended meaning is: while text is ambiguous, our learned representation must be specific enough to resolve this ambiguity. **As long as the representation can steer the generative process to successfully recover the *original image* (given a fixed initial noise), it qualifies as the target representation we seek.** We have revised the text to clarify that our goal is to ensure the generation outcome aligns with the original image.
>
>
> > Major Weakness 4: "line 267-269: This does not explain why only low level features are not the right way to go. Additionally, in fact [2] show that if post-processing is applied during training (getting rid of some low-level features), the model generalizes better, which seems contrary to the hypothesis being presented here. I believe the paper would benefit by being more clear here."
> ***
>
> **Response:**
>
> We thank the reviewer for the comment. We respectfully clarify that lines 267–269 do not imply that “only low-level features are not the right way.” Instead, our argument is that relying solely on high-level features is not comprehensive, since low-level features are indispensable for diffusion model architectures.
>
> Regarding the result in Wang et al., 2020 that removing low-level features improves generalization, this finding is limited to GAN architectures and does not apply to diffusion models. Their evaluation was conducted exclusively on GAN-generated images, leaving the effectiveness of this strategy for diffusion detection untested. Detectors designed for GAN-generated images often fail on diffusion-generated images  because this performance drop reflects the fundamental differences in the forensic traces produced by the two generative architectures [1].
>
> Reliable detection of diffusion-generated images requires incorporating low-level features that capture subtle microstructure and texture-level irregularities. Zhong et al., 2025 [2] show that a diffusion-based low-level feature extractor effectively identifies these fine-grained artifacts, demonstrating the necessity of low-level feature are necessary for diffusion-based deepfake detection.
>
> References:
>
> [1] Corvi, R., Cozzolino, D., Zingarini, G., Poggi, G., Nagano, K., & Verdoliva, L., “On the Detection of Synthetic Images Generated by Diffusion Models,” ICASSP 2023.
>
> [2] Zhong, N., Chen, H., Xu, Y., Qian, Z., & Zhang, X., “Beyond Generation: A Diffusion‑based Low-level Feature Extractor for Detecting AI-generated Images,” CVPR 2025.

---

> ### Author Response · Authors · 2025-11-25
> **(3/4) Rebuttal from Authors of Submission12427 to Reviewer S9Kt**
>
> > Major Weakness 5: "Line 305-306 talks about how any representation with high-level and low-level features is not necessarily helpful. This raises doubts on what is the thing that is special about the representations obtained by TOFE. It would be helpful if the authors can conduct experiments to understand this in a deeper manner."
> ***
>
> **Response:**
>
> We thank the reviewer for the insightful comment. The key advantage of **TOFE** is that it achieves **functional fusion** of these features through a **reconstruction-driven optimization process**, rather than relying on a superficial dimensional concatenation.
>
> In contrast to concatenation — which merely places heterogeneous features side by side — **TOFE operates under a unified reconstruction objective** that makes **high- and low-level features jointly necessary** for minimizing the loss. High-level semantics are required to recover global structure, while low-level features are indispensable for reconstructing fine textures and edges. Because neither type of feature alone suffices, the optimization enforces a **tightly coupled representation** in which both information sources interact within a compact latent space. This yields an embedding shaped by **information necessity**, not dimensional expansion, leading to more discriminative and complete features.
>
> Figure 2 illustrates this contrast clearly. When the embedding contains only high-level semantics (cyan border), the reconstruction preserves the coarse layout of the hotel room but fails to reproduce essential fine-grained features—the texture of the bedding becomes blurred, furniture details disappear, and boundary details are inaccurate. This shows that high-level information alone is insufficient for pixel-level fidelity. In contrast, the blue border successfully restores both global structure and local details: the contours of the furniture are precise, textures are retained, and small patterns such as edges and shadows are accurately recovered. This example demonstrates how the reconstruction objective compels TOFE to integrate high- and low-level features into a single, functionally complete representation, whereas semantic-only features cannot satisfy the reconstruction requirements.
>
> Empirically, **TOFE representations achieve high-fidelity reconstruction (PSNR > 30, SSIM > 0.87; Table 5)** and consistently **outperform other features in downstream detection** (Tables 2 and 3), confirming that they provide a more effective integration of high- and low-level features.
>
> >  Major Weakness 6: "The paper currently lacks experiments studying the sensitivity to post-processing. Conducting a sweep over JPEG/Resizing (both upsampling and downsampling), WEBP and blur would make the work more suited for practical applications."
> ***
>
> **Response:**
> We thank the reviewer for this insightful comment. As seen in **Appendix A.7.** The results confirm that our method exhibits high stability against minor perturbations (such as high-quality compression), showing slight performance degradation. Even under severe corruption, the model maintains a certain degree of practical utility. As detailed in the table below:
>
> | Robustness | Level | Avg Accuracy (%) | Avg AP (%) |
> |:---|:---|:---|:---|
> | **JPEG** | Q90 | 97.09 | 99.46 |
> | **JPEG** | Q80 | 81.93 | 95.31 |
> | **JPEG** | Q70 | 66.07 | 75.93 |
> | | | | |
> | **Crop** | 0.9 | 58.18 | 71.54 |
> | **Crop** | 0.7 | 56.85 | 69.63 |
> | **Crop** | 0.5 | 77.02 | 80.45 |
> | | | | |
> | **WebP** | Q95 | 90.92 | 98.41 |
> | **WebP** | Q90 | 72.89 | 81.04 |
> | **WebP** | Q85 | 68.44 | 79.27 |
> | | | | |
> | **Noise** | 5 | 66.33 | 71.35 |
> | **Noise** | 10 | 58.08 | 65.80 |
> | **Noise** | 15 | 46.29 | 59.58 |
> | | | | |
> | **Blur** | 3 | 60.10 | 62.82 |
> | **Blur** | 2 | 57.15 | 58.20 |
> | **Blur** | 1 | 56.03 | 80.34 |
>
> >  Minor Weakness/Suggestion 1: "Line 41: If the main task is fake image detection, what is the downstream task."
> ***
>
> **Response:**
> We thank the reviewer for the clarification. Our intent was to refer to the downstream task following feature extraction, which is detection. We will revise the text to avoid this confusion.
>
>
> >  Minor Weakness/Suggestion 2: "Line 42: Ojha et al is designed for universal fake image detection (they test on GANs as well, not only diffusion)"
> ***
>
> **Response:**
> We thank the reviewer for the clarification. We agree that Ojha et al. is designed for universal fake image detection (including GANs), and we will revise the text.
>
> >  Minor Weakness/Suggestion 3: "Line 224-225: DDIM is not the only way to perform denoising (DDPM + other solvers such as PNDM would also work)."
> ***
>
> **Response:**
> We thank the reviewer for the clarification. We agree that DDPM and other solvers are effective denoising methods, and we will revise the text to reflect this.

---

> ### Author Response · Authors · 2025-11-25
> **(4/4) Rebuttal from Authors of Submission12427 to Reviewer S9Kt**
>
> >  Minor Weakness/Suggestion 4: "Line 42: DIRE is cited as a feature extraction method, but it is reconstruction-based and it would be better if it was distinguished from other feature extraction methods)."
> ***
>
> **Response:**
> We thank the reviewer for this point. We agree that DIRE should be clearly distinguished from pure feature extraction methods. While DIRE is primarily reconstruction-based, it still relies on a ResNet model for feature extraction. We will clarify this distinction in the revision.
>
> >  Minor Weakness/Suggestion 5: "Sec 4.2.2 is titled Problem Definition, but it talks about the particular algorithm developed by the authors. Problem Definition should talk about the problem which the authors are trying to solve."
> ***
>
> **Response:**
> We thank the reviewer for pointing this out. We agree that Section 4.2.2 currently describes our specific algorithm. We will revise the section title to accurately reflect its content in the revision.
>
> >  Minor Weakness/Suggestion 6: "Line 62-63: distinguishing between real and generated images within the image domain is becoming increasingly challenging. This claim is not entirely true. While generalizing to other generators/unseen post-processing operations remain challenging, usually trained classifiers are able to discriminate between real and fake distributions successfully. I believe this statement needs to be worded with further clarity, since it is critical to the design choice."
> ***
>
> **Response:**
> We thank the reviewer for requesting greater clarity in this statement. Our intended challenge focuses on Diffusion Models. While traditional detectors successfully rely on artifact features left by GANs in the image domain, the significantly improved image generation performance of Diffusion Models means their generative mechanism significantly weakens the image-domain artifacts traditionally utilized by detectors [1]. Given this, the discriminative signal in the image domain is diminished. We believe it is **worthwhile to explore other domains to extract more effective features**, which is the core driver of our design choice. We will revise Lines 62-63 to clearly articulate this technical rationale.
>
> References:
>
> [1] Corvi, R., Cozzolino, D., Zingarini, G., Poggi, G., Nagano, K., & Verdoliva, L., “On the Detection of Synthetic Images Generated by Diffusion Models,” ICASSP 2023.
>
> > Question 1: "Is the Image being DDIM inverted using the unconditional diffusion model? If so why?"
> ***
>
> **Response:**
> We used the **conditional** diffusion model for DDIM inversion, as shown in Section 4.2.3 and Figure 3 in the main text.
>
> > Question 2: "Related to above, it seems to me like the DDIM sampling is done from the unconditionally inverted latent. Why is the embedding being optimized to align with the unconditional trajectory. One possible solution to the same could be the null-text embedding (which would mimic the original trajectory). Please clarify the motivation behind this method."
> ***
>
> **Response:**
> We appreciate your insightful comment. You are precisely correct: our method uses the **null-text embedding** ($\oslash$) as the starting condition for optimization.
>
> We are not using an unconditional diffusion model. The entire process uses the **conditional LDM** and its **classifier-free guidance** mechanism. We optimize the null-text embedding to precisely guide this conditional sampling to match the inverted latent trajectory $\{z_t\}_{t=0}^T$. This procedure is visualized in **Figure 3** and detailed in **Section 4.2.3**.
>
> > Question 3: "Algorithm 1: It starts by saying z_0 is obtained from the image (seems like it comes from the latent encoder). Then it proceeds to say that z_0 is computed from DDIM inversion which is confusing to me."
> ***
>
> **Response:**
> We appreciate you clarifying this detail. To be precise, $z_0$ is derived from the input image $I$ using the latent encoder. We will ensure this is clearly phrased in the final manuscript.
>
> > Question 4: "Line 299-302 talks about extracting features at every time step. However, only T=1 is being used in line 350-351. This design choice would benefit from stronger motivation."
> ***
>
> **Response:**
> We appreciate you raising this point regarding our timestep selection.
>
> Our methodology is broadly applicable for feature extraction at **any timestep $T$** (as described in Algorithm 1 and the surrounding text). To demonstrate this generality, we evaluated performance using both **$T=1$** and **$T=50$**.
>
> We observed that the detection performance remained high and comparable for both timesteps. We chose **$T=1$** for our main results (lines 350-351) primarily for **efficiency**, as it offers the fastest feature extraction time.
>
> The high performance achieved with $T=50$ confirms the **generality** of our feature extraction across the trajectory, and these results are provided in the ablation study in **Section A.4.3** and **Table 7**.

---

### Official Review · Reviewer_ALvA · 2025-11-01

**Soundness:** 3
**Presentation:** 3
**Contribution:** 3
**Rating:** 6
**Confidence:** 4

**Summary:**

The authors explore whether cross-model feature extraction can be instrumental in deepfake detection. They highlight the benefits of using a combination of high- and low-level features for deepfake detection. As a remedy, the authors propose  mapping images into other modalities, such as text, which helps distinguish between the real and fake data in the image domain.  The authors demonstrate their findings empirically.

**Strengths:**

Originality: the paper considers the combination of cross-modal low- and high-level features. It is my understanding that investigating the diffusion-model inversion to extract the text-guided features is a novel and promising idea. Interestingly, it also shows the transferability to the GAN models

Quality: the paper is written well, and seems reproducible. The algorithm description and the justification for the method look correct.

Significance: the paper's significance is in discovering new method to analyse latent features via inversion.

**Weaknesses:**

Novelty: there is a need to contrast this work with the existing ones also looking at the low- and high-level feature extraction (see Question 1).

**Questions:**

1. the idea of combining low- and high-level features for deepfake detection is a known one, and it has been seen before (see, e.g., Liu et al (2022). Of course, it has not been seen in this setting, but it would be good if the authors could contrast between these work and the proposed one from the angle of combining the high- and low-level features.

Liu et al (2022) Cross-Domain Local Characteristic Enhanced Deepfake Video Detection, ACCV

2. The point about transferability (Section 5.4) is very interesting. I would like to ask the authors to say if it is possible to present additional experimental evidence across different non-diffusion architectures, so that we can see if this observation persists.

3. Is it possible to present confidence intervals?

4. Figure 1 presentation can be improved with enlarging the points

---

> ### Author Response · Authors · 2025-11-25
> **(1/3) Rebuttal from Authors of Submission12427 to Reviewer ALvA**
>
> We are sincerely grateful for your valuable feedback and the careful attention you devoted to reviewing our work.  Our point-by-point response to your concerns is provided below:
>
> > Weakness: "Novelty: there is a need to contrast this work with the existing ones also looking at the low- and high-level feature extraction (see Question 1)."
> >
> > Question 1: "the idea of combining low- and high-level features for deepfake detection is a known one, and it has been seen before (see, e.g., Liu et al (2022). Of course, it has not been seen in this setting, but it would be good if the authors could contrast between these work and the proposed one from the angle of combining the high- and low-level features."
> ***
>
> **Response:**
>
> We appreciate your feedback and would like to clarify the main contributions and novelty of our approach. We assert that our proposed TOFE method introduces fundamental differences and significant innovation compared to existing multi-level feature fusion techniques, such as Liu et al. (2022).
>
> ### 1. Fundamental Difference in Feature Extraction Domain: Cross-Modal
>
> The core novelty lies in shifting the detection task from the image domain to the text domain:
>
> * Liu et al. (2022) and Similar Works: These methods perform feature fusion **within the image modality**. For instance, they combine different representations like spatial domain and frequency domain features.
> * TOFE: We employ a **cross-modal** feature extraction paradigm. We convert the image's visual information into a **continuous text embedding** ($\hat{\mathcal{C}}$). This transformation introduces a new, more discriminative feature space for detection, which is crucial for distinguishing highly realistic diffusion-based images.
>
>
> ### 2. Uniqueness of the Fusion Mechanism: Reconstruction-Constrained Encoding
>
> We replace conventional feature combination with a novel optimization process:
>
> * Liu et al. (2022) Mechanism: Fusion often relies on concatenation or implicit learning of image-domain features. Our analysis shows (Section 3) that simple concatenation is insufficient and unstable for this task.
> * TOFE: We achieve feature fusion explicitly by iteratively **optimizing the continuous text embedding** $\hat{\mathcal{C}}$. This process is governed by a **reconstruction constraint**, which minimizes the difference between the target image and the image generated by the optimized embedding. This constraint effectively forces $\hat{\mathcal{C}}$ to unify and encode both **High-level Semantic Information** and **Low-level Fine-Grained Visual Details**. This novel, reconstruction-driven encoding mechanism is the source of TOFE's clear advantage in separating real and fake image distributions.

---

> ### Author Response · Authors · 2025-11-25
> **(2/3) Rebuttal from Authors of Submission12427 to Reviewer ALvA**
>
> >  Question 2: "The point about transferability (Section 5.4) is very interesting. I would like to ask the authors to say if it is possible to present additional experimental evidence across different non-diffusion architectures, so that we can see if this observation persists."
> ***
>
> **Response:**
>
> We appreciate the reviewer's suggestion to further evaluate the transferability of our method. In addition to the GAN-based models presented in the paper, we conducted supplementary experiments on Diffusion Transformer (DiT) and Flow Matching architectures.
>
> **Table 1: TOFE Feature Detection Performance (Transferability Experiment) Across Different Generative Models**
>
> | Model Name       | Generation Method | ACC (%) | AP (%) | R_ACC (%) | F_ACC (%) | F1 (%) | ROC (%) |
> | :--------------- | :---------------- | ------: | -----: | --------: | --------: | -----: | ------: |
> | Dalle3           | Closed-Source     | 96.36   | 93.10  | 98.60     | 81.13     | 84.94  | 98.80  |
> | hunyuan          | Diffusion         | 85.67   | 98.02  | 98.60     | 72.65     | 83.51  | 98.20  |
> | SD3.5            | Flow Matching     | 51.99   | 68.00  | 98.61     | 5.42      | 10.10  | 68.56  |
> | FLUX.1-dev       | Flow Matching     | 50.23   | 59.84  | 98.60     | 1.92      | 3.72   | 64.72  |
>
>
> Our results indicate that TOFE maintains strong transferability across GAN-based models (average ACC $\sim 97$\%) and DiT architectures (e.g., Hunyuan-DiT ACC $\sim 86$\%). This confirms that our feature space effectively captures general discrepancies between real and synthetic images across different backbones (CNNs and Transformers). However, we observe a critical limitation when testing on Flow Matching models (e.g., SD3.5, FLUX.1-dev), where detection accuracy drops to near random chance ($\sim 50$\%).
>
> We attribute this limitation to a fundamental **mismatch in the feature encoding mechanism**. TOFE relies on the Latent Diffusion Model (LDM) manifold to capture discriminative signals, specifically the low-level artifacts and noise patterns characteristic of discrete DDPM/DDIM processes. However, Flow Matching models operate on **continuous probability flows**, producing unique low-level distributions that diverge significantly from the LDM prior. Consequently, the LDM-based encoder is **insensitive** to these architecture-specific cues, resulting in an embedding that fails to capture the fine-grained information necessary for detection. **This finding indicates that the transferability of detection methods to Flow Matching architectures presents unique challenges, and further exploration in this direction is warranted.**

---

> ### Author Response · Authors · 2025-11-25
> **(3/3) Rebuttal from Authors of Submission12427 to Reviewer ALvA**
>
> > Question 3: "Is it possible to present confidence intervals?"
> ***
>
> **Response:**
>
> We appreciate the suggestion. We have calculated the **95% confidence intervals** for our reported metrics using the **bootstrap method** with **1,000 resamplings**. The results indicate narrow intervals across datasets, confirming that the performance stability of TOFE is statistically significant.
>
> **Table 2: Model Performance (Including 95% Confidence Intervals)**
> | **Models** | **Accuracy (%)** | **AP (%)** | **F1 (%)** | **Recall (%)** | **ROC AUC (%)** |
> | :---: | :---: | :---: | :---: | :---: | :---: |
> | **ADM** | 97.28 [96.55, 97.95] | 99.72 [99.58, 99.84] | 97.27 [96.59, 97.96] | 95.98 [94.67, 97.19] | 99.73 [99.59, 99.84] |
> | **PNDM** | 99.09 [98.70, 99.50] | 99.95 [99.91, 99.99] | 99.10 [98.64, 99.49] | 99.59 [99.19, 99.90] | 99.95 [99.90, 99.99] |
> | **IDDPM** | 99.01 [98.55, 99.40] | 99.92 [99.85, 99.98] | 99.00 [98.54, 99.41] | 99.39 [98.88, 99.80] | 99.92 [99.84, 99.98] |
> | **DDPM** | 98.40 [97.73, 99.04] | 99.61 [99.31, 99.84] | 97.46 [96.37, 98.46] | 97.98 [96.58, 99.12] | 99.82 [99.70, 99.92] |
> | **IF** | 98.50 [97.95, 99.00] | 99.87 [99.77, 99.94] | 98.51 [97.94, 98.98] | 98.38 [97.52, 99.11] | 99.85 [99.71, 99.94] |
> | **LDM** | 98.13 [97.55, 98.70] | 99.82 [99.71, 99.90] | 98.15 [97.51, 98.71] | 97.71 [96.75, 98.55] | 99.80 [99.69, 99.89] |
> | **DALLE2** | 97.91 [97.20, 98.67] | 99.38 [98.94, 99.74] | 96.87 [95.76, 97.88] | 96.59 [94.83, 98.03] | 99.52 [98.99, 99.86] |
> | **SDV1** | 97.54 [96.85, 98.20] | 99.76 [99.65, 99.86] | 97.54 [96.84, 98.19] | 96.50 [95.38, 97.57] | 99.76 [99.65, 99.85] |
> | **SDV2** | 96.86 [96.10, 97.60] | 99.59 [99.38, 99.78] | 96.79 [95.98, 97.52] | 95.09 [93.68, 96.34] | 99.50 [99.17, 99.75] |
> | **VQ-Diffusion** | 99.26 [98.85, 99.60] | 99.98 [99.96, 100.00] | 99.25 [98.84, 99.60] | 99.90 [99.69, 100.00] | 99.98 [99.96, 100.00] |

---

### Author Response · Authors · 2025-12-02
**Overall Response**

We sincerely appreciate the reviewers for their thoughtful and constructive feedback. We are encouraged by the positive recognition of our contribution, which can be summarized as follows:

*  The concept of using diffusion model inversion to extract features in the text modality for DeepFake detection is highlighted as a novel idea. (Reviewers `ALvA`, `brHr`)
*  The paper is praised for its originality in effectively combining cross-modal high- and low-level features. (Reviewers `ALvA`, `brHr`)
* TOFE method demonstrates consistent and strong performance, outperforming established baselines on the DIRE dataset. (Reviewers `S9Kt`, `c3dr`, `brHr`)
*  The method shows valuable transferability to non-diffusion architectures like GANs. (Reviewer `ALvA`)
*  The manuscript is well-written, the methodology is justified, and it provides sufficient experimental evidence, suggesting good reproducibility. (Reviewers `ALvA`, `brHr`)

In response to the feedback, we have carefully addressed each of the concerns raised by the reviewers. Below are the main points and supplementary evidence provided:

* We have clarified the novelty by explicitly contrasting TOFE’s cross-modal, reconstruction-constrained fusion mechanism with existing multi-level feature fusion techniques (details in Rebuttal 1/3 to `ALvA`, Response to Question 1). (Reviewer `ALvA`, `S9Kt`)
* We have documented the comprehensive robustness analysis regarding sensitivity to post-processing (JPEG, WEBP, Blur, etc.) to evaluate practical applications, with detailed results presented in Appendix A.7 of the supplementary material (details in Rebuttal 3/4 to `S9Kt`, Response to Weakness 6). (Reviewer `S9Kt`, `brHr`)
* We have provided additional experimental evidence on transferability across different non-diffusion architectures (including DiT and Flow Matching models), complete with analysis of architecture-specific challenges (details in Rebuttal 2/3 to `ALvA`, Response to Question 2). (Reviewer `ALvA`, `brHr`)
* We have documented the robustness analysis against edited images (Qwen-Image-Edit) and adversarial examples (PGD attacks), demonstrating TOFE's capability in challenging real-world scenarios (details in Rebuttal 3/3 to `brHr`, Responses 1 & 2). (Reviewer `brHr`)
* We have provided a detailed explanation on why the reconstruction-driven optimization process enables the functional fusion of features, leading to enhanced downstream detection performance (details in Rebuttal 3/4 to `S9Kt`, Response to Weakness 5). (Reviewer `S9Kt`, `brHr`)
* We have included 95% confidence intervals for all core performance metrics using the bootstrap method, confirming the statistical significance and stability of TOFE’s performance (results presented in Table 2 in Rebuttal 3/3 to `ALvA`). (Reviewer `ALvA`)
* We conducted an ablation study comparing null-text initialization vs. caption-initialized embeddings (BLIP), demonstrating that null-text yields superior MMD, reconstruction quality, and detection performance, providing motivation for our choice (results presented in Tables 2-4 in Rebuttal 2/2 to `c3dr`). (Reviewer `c3dr`)
* We have provided an analysis of efficiency trade-offs and potential optimizations, showing that our exploration of sampling step reduction (T=50 to T=1) significantly improved speed while simultaneously enhancing detection performance, reducing time overhead to an acceptable level (0.607 seconds/image, approximately 4.89x faster than DIRE) and discussing pathways for future real-time optimization (details in Rebuttal 3/3 to `brHr`, Response to Question 3). (Reviewer `brHr`)
* We have provided clarifications on key implementation details, including the motivation for T=1 timestep selection, the use of the conditional LDM, and precise problem definition terminology (details in Rebuttal 4/4 to `S9Kt`, Responses to Questions 1-4). (Reviewer `S9Kt`)

These detailed responses, along with new experimental results and analyses, are fully documented in the point-by-point rebuttal. We believe these additions further demonstrate our contributions and further address the reviewers' concerns.

Once again, we highly value the reviewers' insightful feedback and are open to any further discussions.

---

### Meta-Review · Area_Chair_Q8tg · 2026-01-06

**Summary:**

Although the authors addressed many specific queries and supplemented experiments, the most critical concerns—regarding the depth of the contribution, the fundamental reason for the method's effectiveness, and the fatal flaw in generalizing to modern generative architectures—remain inadequately resolved. This work demonstrates a clever application of diffusion model inversion but fails to provide the novel insight, solid theoretical grounding, or comprehensive empirical validation.

Insufficient novelty of the core contribution and unclear argumentation: The core idea of "fusing high-level and low-level features" has been explored in existing detection literature. While the authors argue that the innovation lies in the "cross-modal fusion mechanism via text embedding optimization," this distinction was not clearly established in the original manuscript, and the rebuttal fails to provide sufficient evidence that this method offers a fundamental advantage or theoretical breakthrough compared to other multi-feature fusion strategies.

Lack of solid theoretical foundation and mechanistic explanation: Reviewers explicitly pointed out that the paper fails to explain why the representations learned by TOFE are uniquely effective. The authors' response merely reiterates the algorithmic process ("functional fusion through reconstruction-driven optimization") without providing a causal mechanistic explanation—specifically, what discriminative signal the text embedding captures that is absent or weaker in image-domain features. This makes the core innovation appear more like an empirical observation lacking a theoretical anchor.

 Severe deficiency in generalizing to modern generative architectures: The supplementary transfer experiments in the rebuttal reveal a critical weakness: TOFE's performance plummets to near-random levels on state-of-the-art Flow Matching models (e.g., SD3.5, FLUX.1-dev). This severely undermines its claim as a universal cross-architecture solution and indicates that the method is overly sensitive to the specific generative model priors (LDM manifold) it relies on, demonstrating insufficient robustness for practical applications.

 Fundamental weaknesses in methodology and argumentation: The reviewer who assigned a clear "reject" rating identified multiple major issues with the paper's motivation, claims, and experimental design. These include unclear problem definition, overstated claims about the difficulty of image-domain detection, and a lack of sufficient ablation studies to isolate the components truly driving performance. The authors' point-by-point response did not fundamentally alter this reviewer's likely judgment that the core argumentation remains flawed.

**Reviewer Concerns:**

Concerns Addressed:
Experimental Supplements: Expanded transferability experiments (DiT, Flow Matching models), added statistical confidence intervals, specified the definitions of high- and low-level features, supplemented post-processing robustness analysis, clarified technical details, conducted initialization comparison experiments, evaluated adversarial examples, and performed efficiency analysis.
Explanation of Principle: Explained the method's effectiveness from three perspectives: the nature of the embedding, information representation, and the optimization objective.

Outstanding Core Concerns:
Fundamental Novelty Remains in Doubt: The argument that its "cross-modal fusion" constitutes a conceptual breakthrough sufficient for publication, rather than an incremental improvement over existing "multi-feature fusion" ideas, remains insufficient.
Lack of Mechanistic Explanation: It still fails to answer what specific, unique discriminative signal the text embedding captures that is superior to that in the image domain. The explanation remains at the descriptive level of "what it is," without revealing the causal mechanism.
Major Practical Deficit Exposed: The method completely fails on Flow Matching models, indicating its effectiveness heavily relies on LDM/DDPM priors. This severely undermines the claim of generalizability and exposes a critical limitation.
 Fundamental Weaknesses in Argumentation Unchanged: Although technical responses were provided, the reviewer's fundamental doubts about the paper's motivation, claims, and argumentative rigor remain. The inherent weaknesses in the paper's core narrative and evidentiary support have not been repaired.

**Reviewer Scores:**

ALvA (Original Score: 6, Marginally Above Threshold)
Their concerns (novelty contrast, transferability evidence, confidence intervals) were directly and satisfactorily addressed in the rebuttal through new experiments and clear explanations. However, their initial positive view of the method's originality might become more cautious due to the catastrophic failure on Flow Matching models revealed in the rebuttal, leading them to maintain their original borderline position.

S9Kt (Original Score: 2, Reject)
The fundamental weaknesses this reviewer identified in the paper's motivation, claims, and experimental basis were not resolved in the rebuttal. The authors' response focused on technical details but failed to address the core critique: the lack of a fundamental, causal explanation for why TOFE's representations are uniquely effective. The failure on Flow Matching models would further validate their skepticism about the method's robustness. Their high confidence suggests a firm stance unlikely to be swayed.

c3dr (Original Score: 6, Marginally Above Threshold)
While their main concerns (clarification of t-SNE/MMD issues, initialization ablation) were addressed, the primary reason for a score drop is the new critical information revealed in the rebuttal: the method's near-complete failure on Flow Matching models. This information severely undermines the paper's claims about effective cross-architecture feature extraction and could shift them from weak acceptance to rejection.

brHr (Original Score: 4, Marginally Below Threshold)
They explicitly raised concerns about practical scenarios, generalization to new models, and theoretical support. While the rebuttal addressed these points, it also exposed the method's most significant flaw: its failure on Flow Matching models. This directly and negatively addresses their concerns about "real-world scenarios" and "state-of-the-art generative models." Given their initial score was already below the threshold, this evidence of a critical limitation would solidify their reject stance.

Full discussion would harden the consensus toward rejection. The most significant outcome of the rebuttal was not just the responses themselves, but the revelation of a major, previously unknown weakness. This new information is detrimental to the paper's claims. It would likely cause the two originally borderline-positive reviewers (ALvA, c3dr) to become more cautious or negative and strongly reinforce the negative positions of S9Kt and brHr. The final consensus would shift from an initially mixed/weakly positive impression to a clearer agreement that the paper has unresolved, fundamental issues regarding generalizability and explanatory depth, warranting rejection.

---

### Decision · Program_Chairs · 2026-01-26

Reject